# Unaddressed non-energy use in the chemical industry can undermine fossil fuels phase-out

Marianne Zanon-Zotin[1,2], Luiz Bernardo Baptista [1,3], Rebecca Draeger [1], Pedro R. R. Rochedo [4] ✉, Alexandre Szklo [1] & Roberto Schaeffer [1]

Around 13% of fossil fuels globally are used for non-combustion purposes. Fossil fuel processing plants, such as petroleum refineries, exhibit interdependent material and energy system dynamics, making the transition away from fossil fuels in energy systems more challenging without addressing the non-energy outputs. This study explores the future role of fossil fuels for non-energy purposes in climate-stringent scenarios with restrictions on alternative feedstock availability, focusing on the primary chemicals sector. Using a global integrated assessment model with detailed refining and primary chemicals sectors, findings across various scenarios reveal that up to 62% of total feedstock use in the chemical sector could be provided by alternative sources by 2050. This would require significant scale-up in biomass utilisation and carbon capture technologies. Annual $CO_2$ emissions from the chemical sector could be reduced to as low as $-1$Gt $CO_2$ by the same year if carbon storage in non-recycled and non-incinerated bioplastics is accounted for.

In the 2023 United Nations Climate Change Conference (COP28) Global Stocktake (GST), nearly 200 parties agreed on "transitioning away from fossil fuels in energy systems, (...), so as to achieve net zero by 2050 in keeping with the science". Around 30 Exajoules (EJ), or 13% of the world's total fossil fuel production, are used for non-energy purposes. Of this, two-thirds are used as feedstocks for primary chemicals production[1,2], which contributes to approximately 1 $GtCO_2yr^{-1}$ of emissions. This earns the sector a dual role as hard-to-abate and hard-to-defossilize in mitigation pathways. Yet, there is no reference to the non-energy use of fossil fuels in the GST, revealing that key material-energy links remain unaddressed by global climate policy.

With the demand for fossil fuels in energy systems projected to decrease[3], the materials systems will also be affected primarily because fossil fuel plants simultaneously co-produce fuels and feedstock for materials production. Naphtha, the main feedstock used to produce high-value chemicals (HVCs) (i.e., ethylene, propylene, butadiene and aromatics), is inexpensively co-produced with diesel, gasoline and aviation fuels. Refinery units directly co-produce propylene[4] and aromatics[5] while producing fuels. Natural gas (for example, in North America and the Middle East) and coal (mainly in China and South Africa) are also key feedstocks for producing methanol[6] and ammonia[7].

Meanwhile, the last greenfield fuel-oriented petroleum refineries are expected to be those under construction[8] as new refining capacity presents increasingly higher petrochemical integration[9]. Nevertheless, fuel-oriented refineries are projected to remain operating after mid-century given the long-lived and capital-intensive nature of their assets.

Simultaneously, the demand for materials, as well as its feedstocks, are expected to grow[10]. These materials include but are not limited to synthetic polymers, chemicals, fertilisers and other non-energy oil products such as lubricants, asphalt and solvents[11]. This may be partly attributed to the energy transition itself in the form of the materials requirements for timely and at-scale deployment of

[1]Centre for Energy and Environmental Economics (Cenergia), Energy Planning Program (PPE), COPPE, Universidade Federal do Rio de Janeiro, Rio de Janeiro, Brazil. [2]Copernicus Institute of Sustainable Development, Utrecht University, Utrecht, The Netherlands. [3]PBL Netherlands Environmental Assessment Agency, The Hague, The Netherlands. [4]Research and Innovation Center on CO2 and Hydrogen (RICH Center) and Management Science and Engineering Department, Khalifa University, Abu Dhabi, United Arab Emirates. ✉e-mail: pedro.rochedo@ku.ac.ae

renewable energy technologies such as lightweight car parts, solar panel components, wind turbine blades[12], and fertilisers for biomass production[13].

This shift in oil products' demand and supply patterns raises essential questions about the future of the symbiotic relationship between fossil fuels and primary chemicals in a world striving to achieve stringent climate targets. Many studies have addressed long-term strategies to reduce emissions of primary chemicals to reach climate targets based on fuels switch for process heat generation, feedstock substitution, carbon capture, utilisation and storage (CCUS) and circular (bio)economy. These strategies have been thoroughly analysed in the literature[14–19] individually or in combination.

Despite the decreasing demand for fossil fuels in energy systems due to climate policies[3,20,21], the co-production of energy, feedstocks and chemicals in petroleum refineries and implications to energy use and emissions pathways across sectors remain largely unexplored[22]. Aiming to fill this gap, this article investigates the role of the chemical sector in a global net-zero strategy. Our hypothesis is that the chemical sector, while being both hard-to-abate and hard-to-defossilize, can offer strategic contributions to deep decarbonisation, both sectoral and systemically. An integrated assessment perspective is essential to understand: 1) How the integration of a growing chemical sector with the oil refining sector affects fossil fuels phase-out and decarbonisation of chemicals; and 2) The competition for $CO_2$ (i.e., CCUS), hydrogen and biomass for feedstock substitution with other mitigation measures across sectors that also rely on these resources to reduce emissions. To address this, we use the Computable Framework For Energy and the Environment model (COFFEE), a global integrated assessment model (IAM) based on the MESSAGE framework that accounts for various types of oil qualities and refinery typologies, along with an explicit representation of primary chemicals[21,23]. To our knowledge, none of the Intergovernmental Panel for Climate Change (IPCC) scenarios in the Sixth Assessment Report (AR6) assessed the role of fossil fuels as feedstocks for primary chemicals in detail while assessing refinery activity and co-production[24]. Questions that remain unaddressed, particularly from a global integrated assessment perspective, are the supply of materials co-produced in refineries with decreasing utilisation factors, the availability of alternative feedstocks in the required scale and speed, the final use of biomaterials and whether and for how long they store biogenic carbon. Beyond sector-specific net-zero targets, an integrated perspective is essential for understanding the nexus between energy, materials, emissions, land use and carbon across different sectors, including primary chemicals.

With a focus on supply-side mitigation measures, we explore global technological pathways, carbon feedstocks, energy use and direct emissions scenarios for the primary chemicals industry in scenarios considering Implemented National Policies (NPi) and climate policies aiming at limiting global average temperature increase to below 1.5 °C above pre-industrial levels (1.5C) by 2100. We also test a set of restrictions to limit alternative feedstock availability throughout the century in different pathways aligning with a below 1.5 °C scenario, namely: (i) the assumption that biogenic carbon is stored in unrecycled/unrecovered biomaterials is turned off (MNEToff); (ii) limited scale-up of global carbon capture and storage (gCCS), which also affects the availability of $CO_2$ as carbon feedstock; (iii) limited global biomass availability (PBIO), which affects both energy and non-energy applications and (iv) a combination of all restrictions above (all). We further discuss how these constraints impact and are influenced by the energy sector.

Our study demonstrates that the timely and scaled supply of alternative carbon feedstocks to the chemical sector is crucial for meeting climate goals and phasing out fossil fuels. Scenarios with limitations on biomass supply, carbon capture and storage (CCS) deployment, and carbon storage in biomaterials require earlier and more extensive climate action, and show an additional reduction of at least 6 $GtCO_2$ per year by 2030 compared to the 1.5C scenario. The competition between mitigation and feedstock substitution is evident, as restrictions on bio-based feedstocks and CCS lead to a higher reliance on fossil fuels for producing primary chemicals, prolonging oil use throughout the century. However, we also find that the chemical sector can adopt strategies to ease the burden on other sectors by reaching net removal levels as low as −1$GtCO_2yr^{-1}$ by 2050 if the right policies and conditions are established.

## Results
### Global $CO_2$ emissions and fossil fuel use
Our results show that scenarios with stricter restrictions on global biomass availability, deployment of CCS, and biogenic carbon storage (or material net – MNET) require taking climate action faster, sooner and on a larger scale, as demonstrated in Fig. 1a. Global $CO_2$ emissions in the scenario with all restrictions must be reduced by at least 6 $GtCO_2$ per year more than in scenario 1.5C by 2030 to stay within the $CO_2$ budget. This illustrates that CCS and biomass availability are critical in mitigation across all sectors, including but not limited to the primary chemicals. These restrictions also explain the difference between achieving global net-zero $CO_2$ emissions around 2050 or 2060.

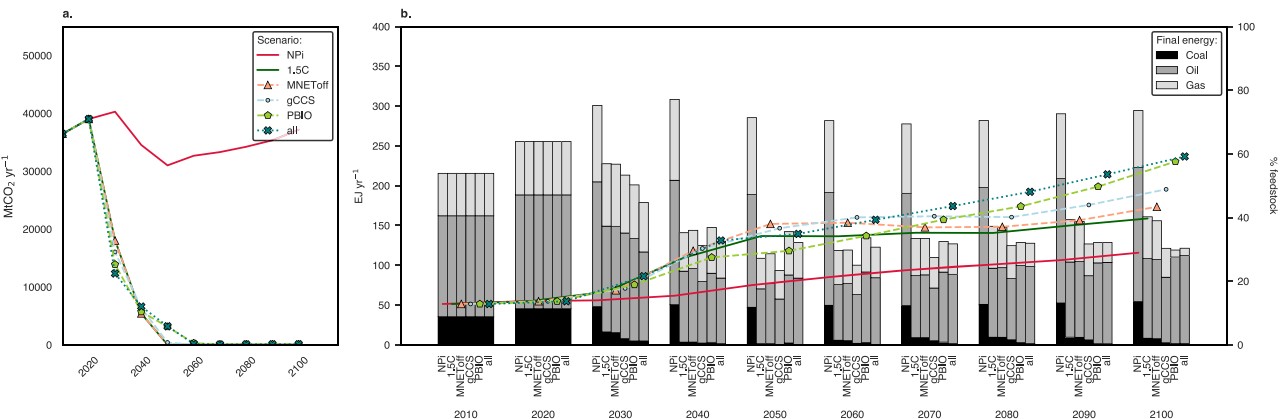

**Fig. 1 | Long-term global $CO_2$ emissions pathways, fossil fuels use and feedstock share across scenarios and sensitivities analysed. a** Global $CO_2$ emissions pathways (2010–2100). **b** Global fossil fuels use (left axis, represented by the stacked bars) and its share used for feedstock purposes (right axis, represented by the lines with scenario markers) (2010-2100). EJ exajoules, NPi Implemented National Policies; 1.5 C: carbon budget consistent with limiting global warming to 1.5 °C; gCCS: a 1.5C scenario with restrictions on global CCS deployment; PBIO: a 1.5 C scenario with constraints on global primary biomass use; MNEToff: a 1.5C scenario that turns off the assumption of biogenic carbon storage in materials; and all: a comprehensive 1.5C sensitivity scenario incorporating all the above-mentioned restrictions. Source data are provided as a Source Data file.

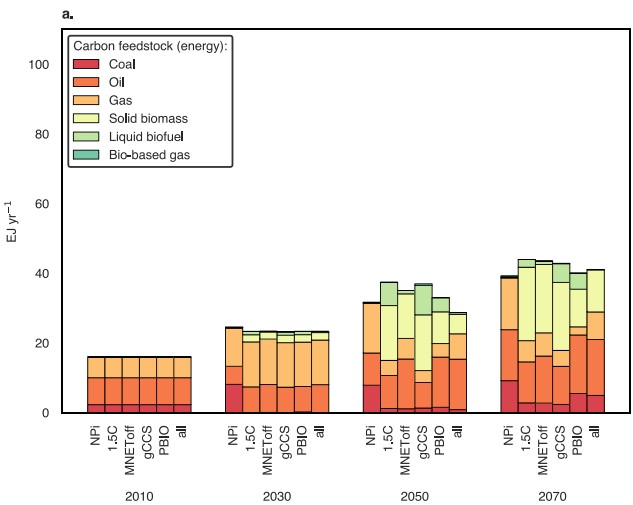
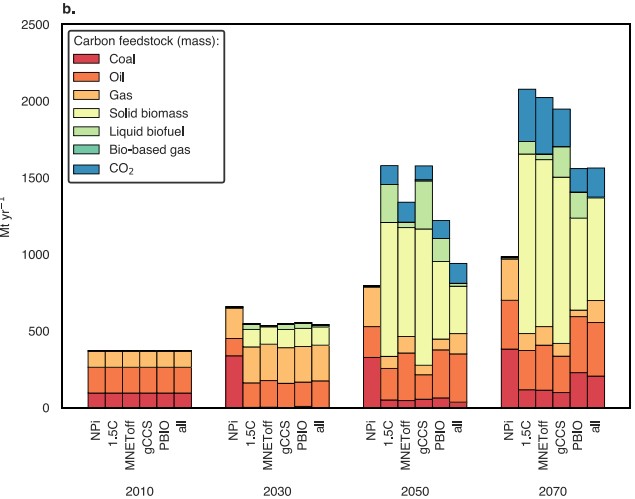

**Fig. 2 | Carbon feedstock sources pathways for primary chemical production.**
**a** Carbon feedstock in energy terms. **b** Carbon feedstock in mass terms. Supplementary Figs. 13 and 14 show the regional pathways for biogenic storage in biomaterials and primary biomass use across R5 regions, respectively. While Asia and the OECD90 + EU have similar biomass use pathways, the Asian R5 region uses up to three times more for non-energy purposes based on the biogenic carbon storage results compared to the OECD90 + EU by 2050. EJ exajoules, NPi Implemented

National Policies; 1.5C: carbon budget consistent with limiting global warming to 1.5 °C; gCCS: a 1.5C scenario with restrictions on global CCS deployment, PBIO: a 1.5C scenario with constraints on global primary biomass use; MNEToff: a 1.5C scenario that turns off the assumption of biogenic carbon storage in materials; and all: a comprehensive 1.5C sensitivity scenario incorporating all the above-mentioned restrictions. Source data are provided as a Source Data file.

While all mitigation scenarios present a similar trajectory to achieve 1.5 °C below pre-industrial levels, we identify distinct fossil fuel use trends, especially after mid-century (Fig. 1b). By 2050, we see that fossil fuels use is reduced from 2020 levels in all of the 1.5 °C scenarios: coal use is reduced by 95-99%, gas by 19-47% and oil use by 40-60% (Fig. 1b).

On the other hand, oil use shows less potential for reduction by 2100 (22-42%) when compared to gas (22-87%) and coal (82-97%). The resurgence in coal use observed after mid-century aligns with the IAM scenario literature and is attributed to the combining effects of: (1) A rising energy demand in regions projected to experience significant population and affluence growth post-2050 (i.e., Africa, Southeast Asia and India), (2) Escalating costs of oil and gas, and (3) Coal use transitioning to include carbon capture[3].

At the same time, the output of fossil fuels transitions away from combustion to non-energy purposes in the NPi scenario, and this trend intensifies in mitigation scenarios. The share of fossil fuels use as non-energy increases from 14% in 2020 to 30-38% in mitigation scenarios by 2050; by 2100, these values further increase to 40-59%. In other words, as restrictions arise in the supply of bio-based feedstocks and the scale-up of CCS (i.e., feedstock based on Carbon Capture and Utilisation - CCU), more fossil fuels – particularly oil – will be required to fulfil carbon-based feedstocks. This happens simultaneously with the electrification of passenger vehicles, leading to gasoline oversupply. As gasoline and naphtha share a similar carbon range, gasoline replaces naphtha as feedstock for HVCs. Therefore, oil use persists throughout the 21st century in our results through these two self-reinforcing mechanisms: (i) gasoline-naphtha substitution flexibility and (ii) insufficient expansion of alternative feedstocks to fulfil the growing primary chemicals demand. This combination leads oil refineries to remain operating in our model for longer, thereby producing fuels, chemicals and feedstocks.

## Oil production and refineries' utilisation factors
Refineries are capital-intensive plants with limited flexibility concerning their product outputs[25,26]. They are designed to optimise profit margins, which are influenced by the quality of crude oil and the product slate required by consumers. Typically, the complexity of refinery operations increases when the crude input shifts from sweet

and light to sour and heavy grades, which requires advanced processing units such as Hydrocracking. The product slate is primarily focused on gasoline and diesel but may also extend the production of specialised products like aviation fuels, petrochemical naphtha and lubricants, depending on market demands.

Our results highlight three trends in the refining sector in mitigation scenarios, which intensify in the more restrictive scenarios. Firstly, the refining sector shrinks in capacity throughout the century (see Supplementary Fig. 10); its utilisation factor also reduces until 2050. Total refining capacity in terms of oil input reduces from ~191 EJ.yr⁻¹ in 2010 to 103 (1.5 C) −132 (all) EJ.yr⁻¹ in 2050 and 44 (gCCS) − 77 (all) EJ.yr⁻¹ in 2070. The upper limits are set by the scenario all, not NPi, indicating that highly restricted scenarios foresee a greater demand for oil. Furthermore, the utilisation factors drop from ~70% in 2020 to 30-50% in mitigation scenarios by 2030 and then increase to 100% in 2070 once the new capacity is built. This follows the dynamics of capital optimisation to maximise long-term value of newly built capacity considering energy and non-energy demands.

Secondly, greenfield refineries become more complex. While old and inefficient refinery capacity dies, new capacity is built based on the Hycon technology, the most complex refinery typology in the COFFEE model. It represents a refinery with hydrocracking and residue hydro desulphurisation units, thus allowing for different oil qualities while achieving higher conversion rates.

Thirdly, greenfield refineries have higher integration with petrochemicals. Although refinery-sourced chemicals are reduced by 49-77% in 2050 in mitigation scenarios (Supplementary Fig. 10), the refinery capacity that survives after mid-century is up to 15% more integrated to petrochemicals than current figures.

## Carbon feedstock sources in a carbon-constrained world
Pathways for carbon feedstocks in primary chemical production are presented in energy (Fig. 2a) and mass (Fig. 2b) terms. By that, we aim to compare fossil and bio-based hydrocarbons, which can be used as either energy or feedstocks, and $CO_2$, which results from carbon capture in the model and is not an energy carrier. Results indicate that mitigation scenarios see an increase in liquid and solid biomass use in the chemical sector, reaching 6–25 EJ.yr⁻¹ by 2050 and 12–25 EJ.yr⁻¹ by 2070. To a lesser extent, $CO_2$ also becomes relevant as a feedstock,

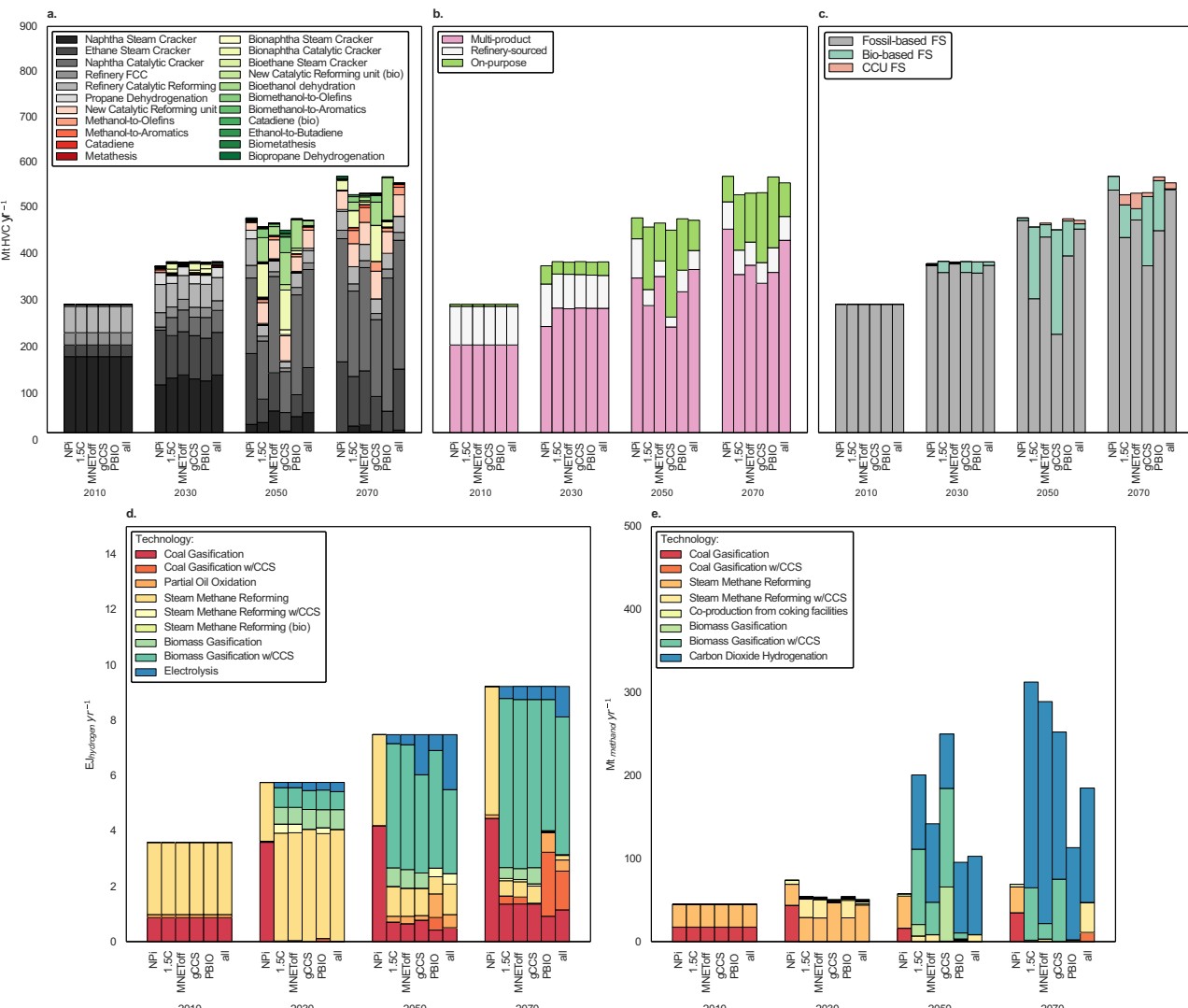

**Fig. 3 | Technology pathways for high value chemicals (HVCs), ammonia and methanol.** HVC production is illustrated in three approaches, being (a) technology split, (b) technology typology and (c) carbon source. In the legend box for plot **a**, colours relate to the feedstocks used as inputs: grey: fossil-based; red: CCU (i.e., CO₂ captured from bio-based or process emissions sources as well as from Direct Air Capture) or fossil-based; green: bio-based. Demands for ethylene, propylene, butadiene, and Benzene, Toluene, and Xylenes (BTX) are static, but ethylene can be used as an intermediate for propylene and butadiene via Metathesis and Catadiene (Supplementary Method 2), thus explaining different production levels across scenarios. **d** Technology split for hydrogen in ammonia production. Ammonia demand is static. **e** Methanol production (only non-energy). Methanol long-term demand is composed of both a static demand and an ancillary demand as an intermediate for ethylene, propylene and BTX via MTO and MTA. This explains the increased demand after mid-century. EJ exajoules, NPi implemented National Policies; 1.5C: carbon budget consistent with limiting global warming to 1.5 °C; gCCS: a 1.5C scenario with restrictions on global CCS deployment; PBIO: a 1.5C scenario with constraints on global primary biomass use; MNEToff: a 1.5C scenario that turns off the assumption of biogenic carbon storage in materials; all: a comprehensive 1.5C sensitivity scenario incorporating all the abovementioned restrictions; FCC: fluidised catalytic cracking; FS feedstock, CCU carbon capture and utilisation, CCS carbon capture and storage. Source data are provided as a Source Data file.

reaching up to 368 Mt.yr⁻¹ by 2070. Interestingly, the most significant deployment of CCU is observed in scenarios where biogenic carbon storage is turned off (MNEToff and all), supporting studies that find that the climate advantages of biomass conversion to materials are somewhat contingent on this assumption[27–29]. Moreover, a comparative analysis of the two plots reveals that feedstock substitution requires more significant mass input to produce an equivalent basket of products. This highlights the variance in fossil and alternative feedstock conversion yields, which have implications for transportation logistics (Supplementary Fig. 6 presents the mass of carbon embedded in feedstocks).

Fossil fuels used as feedstocks in mitigation scenarios achieve reductions of up to 62% by 2050, respective to NPi. This is quite ambitious considering challenges such as developing new supply chains and transportation logistics for alternative feedstocks. At least 500Mt of primary chemicals capacity is <20 years old, and massive investments in fossil-based capacity have been made in the past ten years, particularly in China and Southeast Asia[30,31]. Significantly, scenarios that present reduced petroleum use as carbon feedstock are compatible with reduced oil production (Supplementary Fig. 8).

The upper bound of feedstock substitution (62%) refers to the gCCS scenario, indicating a higher opportunity for Bioenergy with Carbon Capture and Storage (BECCS) and materials use in long-term applications. The lower bound (28%) refers to the scenario all, reflecting the increased need for oil when alternative feedstock availability is constrained. However, as Supplementary Fig. 9 shows, reminiscent oil demand is met primarily with light and medium sweet oil.

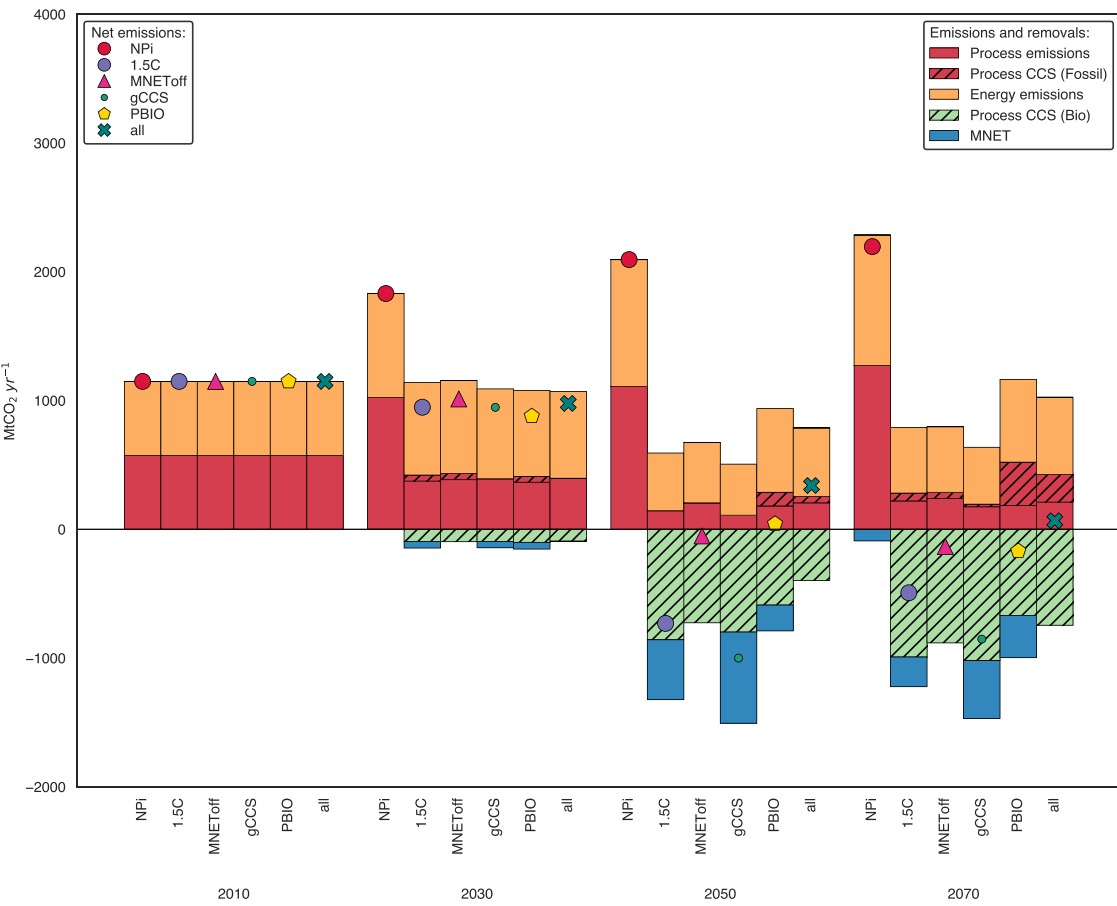

**Fig. 4 | Global direct CO₂ emissions from the chemical sector.** Includes direct (Scope 1) emissions from primary chemicals (explicitly modelled in the COFFEE model) and other chemicals (implicitly modelled in COFFEE) (see Supplementary Method 2). Symbols represent net CO₂ emissions in each scenario. Energy and Process emissions refer to CO₂ emissions resulting from combustion and from chemical reactions not related to energy use, respectively. In this work, we consider emissions from hydrogen production for ammonia synthesis as process emissions. Emissions from incineration and mechanical recycling are also regarded as Process Emissions. 'Process CCS (Fossil)' reflects the reduction in emissions achieved through the CCS of fossil feedstock used to produce ammonia and methanol. 'Process CCS (Bio)' refers to the removal of CO₂ emissions achieved through the CCS of bio-based feedstock used to produce ammonia and methanol. MNET refers to biogenic carbon storage in long-lived materials. Supplementary Figure 12 presents regional pathways for CO₂ emissions in the chemical sector across R5 regions. NPi Implemented National Policies; 1.5C: carbon budget consistent with limiting global warming to 1.5 °C; gCCS: a 1.5C scenario with restrictions on global CCS deployment; PBIO: a 1.5C scenario with constraints on global primary biomass use; MNEToff: a 1.5C scenario that turns off the assumption of biogenic carbon storage in materials; all: a comprehensive 1.5C sensitivity scenario incorporating all the abovementioned restrictions. CCS: Carbon Capture and Storage. Source data are provided as a Source Data file.

These qualities of crude oil have lower emissions and yield a higher output of products per input of raw material[32–34].

When only CCS is restricted, feedstock substitution is prioritised as a mitigation measure in the chemical sector. As Fig. 2d, e shows, biomass gasification with CCS reaches its highest potential in ammonia and methanol production precisely when CCS is restricted globally. This indicates that, under constraints, BECCS is prioritised in the chemical sector.

### Technology pathways and resource allocation

Our results show that steam cracking (SC) remains the leading technology for producing HVCs. Ethane SC increases from 9% to 25% of HVC production in all mitigation scenarios between 2010 and 2030, particularly in gas-rich regions such as the Middle East and the United States. Ethane SC has a higher selectivity towards ethylene over other HVCs, compared to Naphtha SC (Supplementary Fig. 2). As a result, switching from naphtha to ethane as a feedstock leads to propylene and aromatics gaps in demand/supply ratios.

Furthermore, as the passenger transport sector gradually electrifies, refineries reduce their utilisation factors and their HVC output

(Fig. 3b). This results in a gasoline surplus, which becomes cheaper and finds market in the primary chemical sector through Naphtha Catalytic Cracking (NCC) (32–52% of HVC market share by 2070). NCC has a more balanced ethylene/propylene ratio than conventional Naphtha SC[35] (Supplementary Fig. 2) – and catalytic reforming units. In scenario all, the share of electricity use in the transport sector reaches its highest level (37% on final energy use basis considering passenger and freight transport in 2070, see Supplementary Figs. 16 and 17) as biofuels reduce their importance for BECCS given CCS and biomass constraints.

Moreover, reduced HVC output from refineries makes the propylene and aromatics gap more pronounced. NCC deployment is not sufficient to bridge these gaps alone, and on-purpose routes such as Propane Dehydrogenation (PDH), Metathesis (MTT) and Methanol-to-Olefins (MTO) see substantial growth (Fig. 3b).

Furthermore, bioethanol is also diverted from an internal combustion engine fuel to ethylene production. Interestingly, the highest level of ethylene from ethanol is achieved in the PBIO scenario, where biomass use is restricted but not CCS. This means that the model prioritises the limited availability of biomass in the ethanol industry,

finding an opportunity to produce ethanol with CCS and later store biogenic carbon in biomaterials (Supplementary Fig. 8 presents detailed results on technology pathways for each HVC).

To put it concisely, HVC production remains dependent on fossil fuels throughout the century in mitigation scenarios and across all HVCs. However, scenarios 1.5C and gCCS show that 35% and 51% of HVCs are produced with bio-based feedstocks by 2050, respectively (Fig. 3c). As restrictions increase, results for the all scenario seem more and more similar to the results for the NPi scenario, indicating that the lack of availability of alternative feedstocks for HVCs drives the persistent oil use in this sector.

Results on syngas products show that biomass gasification with CCS is favoured for ammonia production due to the potential for BECCS, whereas CCU is preferred for methanol production (Fig. 3d, e). While in ammonia production carbon is largely converted into captured $CO_2$ with high purity, contingent on the efficiency of the capture process, in methanol production, some of the carbon remains in the final product. Therefore, scaling-up biomass use with carbon capture in ammonia production is identified by the model as a source of negative emissions. This can significantly reduce emissions across the overall system both in scale and pace required for reaching 1.5-degrees goals. Nevertheless, water electrolysis coupled with $O_2$ use in oxyfuel routes for capturing $CO_2$ could also be effective, although not assessed in this work. Overall, the use of fossil fuels for ammonia reduces over time but remains relevant, especially in more stringent scenarios.

Furthermore, in the gCCS scenario, we see is that the chemical sector is prioritised for capture across various sectors (see Supplementary Fig. 15). Although the global CCS roll-out declines in 2050 compared to other scenarios, the use of CCS in the chemical sector increases because the model chooses to reduce emissions in the sector with the lowest cost. Note that when biomass and CCS are constrained simultaneously (scenario all), this relative advantage disappears. Similarly, when only biomass is constrained, total CCS remains the same but "Energy CCS – Fossil fuels" increases in relative terms, leading to a greater relevance of Direct Air Capture – or 'DAC'.

### Emission pathways for the global chemical sector

Figure 4 shows the role of the chemical sector in a 1.5 °C world subject to different restrictions. It achieves a net reduction of −0.73 $GtCO_2.yr^{-1}$ in scenario 1.5 C and -1$GtCO_2.yr^{-1}$ in scenario gCCS by 2050. Direct emissions reduction stems mainly from BECCS in ammonia production and biogenic carbon storage, as well as increased efficiency in fossil fuels-based platforms. Not only does the chemical sector not always behave as hard-to-abate from an emissions standpoint, but it can also be a climate asset if bio-based resources are available and the potential of storing biogenic carbon in biomaterials is explored.

However, when all constraints are turned on in the model, the chemical sector remains with residual emissions in 2050, and reaches net-negative emissions only by 2070 (scenario all). This difference of ~1Gt $CO_2.yr^{-1}$ increases the burden on other sectors, which in turn have to extend and anticipate decarbonisation efforts. While accounting for biogenic carbon storage in bioplastics highlights the potential of carbon removal in long-lived applications as a mitigation strategy, it is essential to assess other sustainability dimensions to evaluate its effectiveness as a carbon mitigation strategy fully.

### Discussion

This work contributes to the research field of carbon emissions mitigation in the primary chemicals sector in four ways.

Firstly, we observe that primary chemicals are hard-to-abate but can have a role in promoting systemic decarbonisation. Although decarbonisation technologies are available and could reduce emissions to as low as −1 $GtCO_2yr^{-1}$ by 2050, these reductions heavily rely on the large-scale availability of alternative feedstocks and on accounting for bio-based carbon storage in bioplastics that are neither

incinerated nor recycled. In scenarios where alternative feedstocks are limited and carbon storage in biomaterials is not assumed, achieving net-zero emissions in the chemicals sector by 2070 remains elusive.

Secondly, primary chemicals are hard-to-defossilize, leading to critical implications for decarbonisation of the chemicals sector and beyond. Our findings suggest that ambitious feedstock substitution could reduce fossil fuel dependence by approximately 60% by 2050 globally. However, if carbon capture and biomass use are constrained, the feedstock use profile remains essentially unchanged. This reflects both an opportunity and a challenge. On the one hand, substituting feedstock can significantly reduce refinery utilisation, bringing indirect decarbonisation across sectors. On the other hand, failing to transition away from fossil fuel use as feedstock could inadvertently sustain petroleum refining activities, prolonging fossil fuel reliance and delaying the broader transition away from fossil fuels. This scenario could lead to continued fossil fuel extraction and refining focused on feedstock production, potentially resulting in lower-cost fuel co-production, and assets becoming stranded before investments are amortised. These findings highlight critical implications for investments in fossil fuel assets and underscore the need for a more integrated approach to energy, climate and resource regulation.

Thirdly, transitioning away from fossil fuels will require restructuring within the primary chemicals sector. Steam cracking remained the leading technology for HVC production over the century, largely influenced by the electrification of the transport sector and the resulting diversion of gasoline to the naphtha pool. However, refinery-sourced HVCs will decline due to the shrinking of the refining sector as a result of fossil fuels phase-out policies. On-purpose routes, particularly the ones based on methanol (via MTO, MTA) and ethanol (via BDH, and also influenced by oversupply due to transport electrification) as intermediates, will increase to accommodate feedstock substitutions. The restructuring will also affect methanol and ammonia production; biomass gasification with CCS becomes the preferred route for ammonia, while CCU plays a key role in methanol production. Demand for non-energy methanol in mitigation scenarios is expected to increase two to five times by 2050 compared to NPi. This is primarily due to its importance as an intermediate for HVC production. This shift indicates that solutions are product-specific and every carbon feedstock source will be relevant for achieving climate targets.

Importantly, our scenarios explore a limited set of technologies to understand the role of feedstock supply in achieving net-zero scenarios, assuming unchanged demand patterns for primary chemicals. Given the diverse products and services provided by primary chemicals, detailing and addressing their complexities is challenging. This work aimed to unravel the dynamics of petroleum production and refining within the emissions pathway of primary chemicals by developing drop-in substitution alternatives. Nevertheless, we acknowledge that more is needed to provide a comprehensive view of the potential futures lying ahead for the chemical sector. Historically, energy transitions have redefined systems and triggered material transitions and vice-versa[36]. Hence, transforming the complex network of feedstocks, products and services developed under the petroleum-based economy and thermochemical foundations since the 1950s will require innovative solutions. Beyond phasing-out fossil fuels in energy systems, these include the development of biodegradable chemicals, material substitutions and demand-side measures. While these aspects were not the focus of our assessment, they are crucial considerations for future policymaking.

Fourth and lastly, improving the representation of fossil fuels in IAMs to capture relevant material-energy links allows modellers to reach a higher degree of accuracy in depicting systems integration. Representing incumbent energy carriers set to be phased-out in detail enables modellers to provide realistic but ambitious recommendations to policymakers considering their pervasive use across sectors. In this sense, the scope and scale of

the required transition in the chemical sector to respond to the climate challenges of the 21st century are diverse and in different stages of the supply chain.

A significant albeit deliberate limitation was not including the demand-side or final disposal insights. It is indisputable that these measures hold a highly relevant potential. Stegmann (2022), for instance, found that recycled plastics could make up to 60% of plastics produced yearly by 2050[15]. Chemical recycling (e.g., via pyrolysis, gasification, solvolysis) is also highlighted as an alternative to drastically reduce the future demand for virgin plastics[37,38]. However, we chose to exclude them from this work to analyse the supply-side dynamics in more detail while assuming demand and final disposal shares of plastics as unchanged throughout the century.

Nevertheless, we can draw meaningful conclusions on chemical demand, its final disposal, and its broader role in climate change mitigation. Reducing, Reusing and Recycling/Recovering chemicals are critical circular economy strategies to minimise the investment needed in innovative technologies. Reducing material demand becomes more relevant to the chemical sector than others, given its reliance on fossil fuels as feedstock and the decarbonisation spillover across sectors. On the other hand, if demand reduces, the potential for negative emissions also reduces if feedstock transitions from fossil to bio-based, as was the case for ammonia production in our results.

Furthermore, introducing Crude-Oil-to-Chemicals (COTC) could significantly impact the chemical industry, potentially increasing yields to HVCs or other chemicals to eliminate the problem of material/energy co-production in refineries. However, we did not consider it in our module representation mainly due to a lack of reliable data, which should be covered in future studies. Moreover, we focus on the production of primary chemicals, assuming that the demand for those products remains similar to historical trends and that substitution follows a drop-in logic. In the long run, assuming increased use of biomass and the high weight of oxygen in bio-based feedstocks, oxidised and oxygenated materials such as polyesters or polycarbonates could become the leading platform for plastics. This is also something to be addressed in future work.

Despite these limitations, the significance of these results also lies in quantitatively assessing the climate burden of the carbon lock-in brought about by the increasing economies of scale and scope of fossil fuel resource abundance over decades. Beyond increasing ambition, a change in perspective is needed to embrace the complexity of systems to design climate and resource policy. Without targeting both energy and non-energy purposes in climate agreements such as the GST, the resilience of the fossil fuel industry will continue to be reinforced across decades. Hence, reimagining futures without oil requires targeted modelling and policymaking efforts beyond promoting renewable energy and green hydrogen scale-up. Therefore, policymakers must consider systems interdependencies to design policy frameworks that account for potential synergies and trade-offs between energy and materials systems as well as non-intended outcomes of climate and resource policy.

## Methods

### The Computable Framework For Energy and the Environment (COFFEE) model

The Computable Framework For Energy and the Environment model (COFFEE) is a perfect foresight, linear programming, least-cost optimisation model built within the MESSAGE framework. The model depicts energy and land-use systems in a single framework for eighteen global regions. It is designed to support policymaking by evaluating potential trade-offs and synergies between climate, environmental and energy policies. GHG emissions are accounted for using the IPCC methodology, which includes emissions from fuels, industrial processes, waste and Agriculture, Forestry, and Other Land

Uses (AFOLU), covering greenhouse gases such as $CO_2$, $N_2O$ and $CH_4$. Emissions from biomass production (land-side) for subsequent conversion into energy carriers or materials are accounted for in the AFOLU sector. Additionally, the model incorporates options for carbon dioxide removal, such as land sinks, carbon storage associated with BECCS or DAC, and the use of materials in long-lived applications. Unlike life cycle assessments (LCA) studies that consider cradle-to-grave emissions, COFFEE focus solely on emissions directly associated with major energy, food, and industrial products. Therefore, energy renewable energy sources like solar and wind are considered to provide zero emissions electricity.

COFFEE was one of the five illustrative mitigation pathways (IMPs) highlighted in the last IPCC 6th Assessment Report[24] and detailed information on its structure can be found in ref. 32.

### Oil production and refining sector

COFFEE's resolution of the oil and gas sector is generally higher than that present in other IAMs, which includes the representation of oil qualities, crude trade and fuel-oriented refinery typologies, thus allowing for a better understanding of supply dynamics under climate policy. Refinery typologies can be classified into Existing Topping, Cracking and Hycon, as well as New Cracking and Hycon options. Each type of refinery has three activity modes, which focus on optimising diesel, gasoline, or kerosene (see ref. 32 for a detailed description of those typologies and technoeconomic assumptions).

With this work, we expand this representation to include the regional averages of propylene and BTX output from FCC and CR units, respectively, based on the OGJ Worldwide Refining Survey[39]. We also assumed that greenfield fuel-oriented refinery capacity could increase its HVC output up to 15% to assess potential opportunities for increased petrochemical integration under high-severity operation, based on ref. 40.

### Process representation

Our research expands the scope of the industry sector in COFFEE by explicitly representing the production of HVCs, ammonia and methanol in our model.

For HVC production, we included 12 technologies overall. The reasoning behind our approach was to provide the model with a variety of options for producing each HVC, using a diverse range of carbon feedstock (fossil fuels, biofuels and $CO_2$ via CCU) and industry setups (multi-product or on-purpose routes). Only propylene and aromatics were considered options for refinery co-production, which aligns with prevailing practices and is completely integrated into the refining module's activity. Technologies for HVCs include (naphtha and ethane), steam cracking (SC), naphtha catalytic cracking (NCC), fluidised catalytic cracking (FCC) and catalytic reforming (CR) – both integrated and not integrated to refineries, propane dehydrogenation (PDH), dimerisation (DIM), Catadiene® (CAT), metathesis (MTT), ethanol to butadiene (ETB), methanol-to-olefins (MTO), methanol-to-aromatics (MTA) and bioethanol dehydration (BDH).

For syngas products, the technologies are similar in the concept and feedstock but not in the emissions. The molecule of interest in ammonia production is hydrogen, which will then react with nitrogen. The remaining carbon is converted to $CO_2$, which can be used for urea production or is emitted into the atmosphere. In contrast, hydrogen and carbon monoxide are necessary for producing methanol. Detailed assumptions for emissions accounting can be found in Supplementary Methods 1, 3, 5 and 6. The technologies considered for syngas products are Steam Methane Reforming (SMR), Coal Gasification (CGS), Biomass Gasification (BGS), Partial Oil Oxidation (POX) – only for ammonia, Electrolysis (ELE) – only for ammonia, and Carbon Dioxide Hydrogenation (CDH) – only for methanol. After hydrogen is produced from one of those routes, it is reacted with nitrogen to ammonia via the Haber-Bosch process (HB).

Supplementary Figure 1 illustrates the primary chemicals module and its technologies, and Supplementary Tables 1, 2 and 3 present all techno-economic assumptions. The method for calculating regional long-term demands for each primary chemical is described in Supplementary Method 4.

### Sources of carbon-based feedstock

Carbon-based feedstocks included in COFFEE are categorised as fossil-based, bio-based, and CCU-based, representing products derived from $CO_2$ conversion.

Fossil-based feedstocks include coal, natural gas and products derived from crude oil processing in refineries, including Liquefied Petroleum Gas (LPG), naphtha, and heavy oil. Notably, gasoline can be converted to naphtha and vice-versa.

Bio-based feedstocks include bioethanol and co-products of Fischer-Tropsch Biomass-to-Liquids synthesis (FT-BtL), such as bio-naphtha and bio-LPG. These bio-based feedstocks are derived from processing raw agricultural materials, including sugary, starchy, lignocellulosic and oily crops.

Currently, the only CCU-based feedstock represented is methanol, when produced via the hydrogenation of $CO_2$ captured from diluted (i.e. atmosphere via Direct Air Capture based on $CO_2$ absorption in sodium hydroxide solution) or concentrated sources. Concentrated sources include blast furnaces, power plants, cement kilns and bio- or fossil-based hydrogen production processes, for example. $CO_2$ captured feeds into a regional "$CO_2$ pool" that can be either geologically stored or utilised as feedstock. Supplementary Method 1 presents a more detailed description of the abovementioned feedstocks.

### Scenario development

All of the scenarios are built on the assumptions of the Shared Socio-economic Pathway 2 – the "middle of the road" scenario, which extrapolates historical patterns of social, economic and technological trends throughout the century[41,42]. The assumptions for each scenario are described below and summarised in Supplementary Table 4. Plastics end-of-life and demand patterns were assumed to remain the same throughout the century to allow a more detailed analysis of the supply-side.

### National Policies Implemented (NPi)

The National Policies Implemented (NPi) scenario accounts for the impact of national climate, energy and land policies implemented until 2020, which have long-term implications on carbon dioxide emissions[43,44]. This scenario does not consider additional efforts towards curbing temperature rise beyond what has already been enacted.

### Scenario compliant with 1.5-degree increase limit (1.5C)

Building on the NPi scenario and based on the definitions of the remaining carbon budget from the IPCC Sixth Assessment Report (AR6) Working Group One (WGI) report[45], we create a scenario compliant with the climate targets established under the Paris Agreement: 1.5-Degree Celsius (1.5C) Increase Limit[45]. The budget relative to this scenario is more than 66% consistent with the temperature target (Supplementary Table 4).

We also performed four sensitivity analyses, which allow us to comprehend the model's behaviour when constraints are placed on factors such as CCS deployment (gCCS), biomass availability (PBIO) and assumptions regarding carbon storage in biomaterials (MNEToff), all of which affect either the availability of alternative carbon feedstock ($CO_2$ in gCCS and biomass in PBIO) and the sustainability of bio-based drop-in plastics (MNEToff). These are described below. Both gCCS and PBIO constraints were formulated in scenario protocols within the

framework of model intercomparison initiatives. Specifically, these were undertaken as part of global IAM consortium projects such as the Exploring National and Global Actions to reduce Greenhouse gas Emissions (ENGAGE) and the Next generation of AdVanced InteGrated Assessment modelling to support climaTE policy making (NAVIGATE[46]).

### Scenario with restricted deployment of CCS (gCCS)

Restricted Deployment of CCS scenario (gCCS) envisions a future world where the global deployment of Global Carbon Capture and Storage significantly falls short of expectations. By 2023, CCS facilities captured around 45 Mt.yr$^{-1}$, primarily for Enhanced Oil Recovery. Project announcements foresee a total capacity of 244 Mt.yr$^{-1}$ in the next decade, a figure still below the capture requirement necessary to fulfil the objectives of the Paris Agreement[47]. According to 540 scenarios from IAMs – categories C1, C2 and C3 – in the AR6 IIASA Scenario Database, the maximum deployment of CCS projected for 2030, 2050 and 2100 was 21.0, 30.0 and 38.4 GtCO$_2$.yr$^{-1}$, respectively (Supplementary Fig. 5). This scenario assumes that global CCS achieves 10 Gt.yr$^{-1}$ in 2080.

### Scenario with restricted use of biomass (PBIO)

Based on the AR6 scenario database, biomass use as primary energy ranges from 28.7 to 228.4 EJ.yr$^{-1}$ in 2030, 33.5 to 310.1 EJ.yr$^{-1}$ in 2050 and 41.1 to 530.4EJ.yr$^{-1}$ in 2100 (Supplementary Fig. 6). The Restricted Use of Biomass scenario (PBIO) considers a world where the use of biomass is restricted to below 100 EJ.yr$^{-1}$, based on the high-confidence agreement in the literature found in ref. 48 for the sustainable technical potential of primary biomass production.

### Scenario with biogenic carbon storage turned off (MNEToff)

MNET – or material net – is a constraint specifically designed for this study. The biogenic carbon storage turned off scenario (MNEToff) considers a world where biogenic carbon storage in materials is turned off. The baseline assumption considers that biomaterials act like a carbon sink provided they are not subjected to incineration or degradation[15,28]. We turn that assumption off, thereby assuming that bioplastics can be produced but are not attributed any carbon credits for material storage.

The assumption on biogenic carbon storage currently corresponds to biomaterials in landfills since we consider current regional rates of recycling, landfill and incineration.

### Scenario with all restrictions implemented (all)

The All Restrictions Combined scenario (all) implements all the abovementioned constraints. Not only does the global deployment of CCS develop significantly below expectations, but the use of biomass is also heavily restricted, and carbon credits are not attributed for material storage in biomaterials. This scenario paints a picture of an especially stringent set of restrictions, which helps us to understand the role of the chemical industry in responding to $CO_2$ emissions reduction targets.

## Data availability

Source data are provided with this paper. The data generated by this study are available in the Figshare repository[49]. Source data are provided with this paper.

## Code availability

Although it is not an open-source model, COFFEE is documented on the common integrated assessment model documentation website (https://www.iamcdocumentation.eu/index.php/IAMC_wiki). The code used to generate the figures of this study are available from the corresponding author upon request.

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

## Acknowledgements

M.Z.-Z. was supported by the Coordenação de Aperfeiçoamento de Pessoal de Nível Superior - Brasil (CAPES) by the grant #88887.351681/2019-00. R.D and L.B.B. were supported by the Conselho Nacional de Desenvolvimento Cientifico e Tecnológico (CNPq) by the grants #140453/2022-9 and #141186/2020-8, respectively. This work also received funding from the European Union's Horizon 2020 research and innovation programmes under grant agreement no. 821124 (NAVIGATE) and no. 101056868 (CIRCOMOD). Financial support has been provided by Khalifa University of Science and Technology through the RICH center (project RC2-2019-007).

## Author contributions

M.Z.: conceptualisation, data curation, formal analysis, investigation, methodology, software, visualisation, writing – original draft preparation and writing – review & editing. L.B.B.: conceptualisation, investigation, methodology, software and writing – review & editing. R.D.: data curation, validation and writing – review & editing. P.R.R.R.: conceptualisation, formal analysis, methodology, supervision and writing – review & editing. A.S.: conceptualisation, investigation, project administration, supervision, validation and writing – review & editing. R.S.: conceptualisation, funding acquisition, project administration and writing – review & editing.

## Competing interests

The authors declare no competing interests.
