## [Peer Review File · Nature Communications]

REVIEWER COMMENTS

Reviewer #1 (Remarks to the Author):

This manuscript delves into the intriguing topic of the hurdles associated with transitioning away from fossil fuels as the primary feedstock for chemical production. With detailed models of the complex refinery and base chemical production system, this study enriches existing Integrated Assessment Models (IAMs) by incorporating also the chemical sector. However, there are a number of technical issues in the manuscript, particularly with regard to the assumptions behind, remaining to be addressed or clarified, along with some other grammatical and formatting issues. Hence, before I could recommend it for publication, I urge the authors to address the following aspects:

General comments

1. In the manuscript, the terms “chemicals” and “materials” have been used interchangeably, while in practice, many don’t view it in this way. I would suggest clearly separating them.

Detailed comments

1. L57-58: Perhaps reformulate the statement to make it clearer that it’s the primary chemicals production contributing to ca. 1 Gt CO₂/yr.

2. L73-74: Synthetic polymers, fertilizers, lubricants, asphalt and solvents, etc. are all chemicals. Please revise the sentence here.

3. L120-122: I don’t fully get it here, how come the scenario with all the restrictions has the lowest CO₂ emissions (i.e., overall we can do better without CCS and biomass as feedstocks for the chemical industry?) and how the authors come to the conclusion “that CCS and biomass availability have a critical role in mitigation across all sectors” (i.e., if the previous statement is true, then why CCS and biomass availability is important?). It would be good to provide some elaboration here.

4. Please replace Figure 1 with a high-resolution version and increase the font size to make it readable.

5. L131-132: This seems to be contradictory to L106-107.

6. L138-139: I think it would be good to add some elaboration on the causes here, esp. the increased use of coal.

7. L164-165: How so? Any reference for this statement?

8. L173: Is the increase in capacity utilization due to the phase out of old capacity? What does this increase mean for the chemical industry?
9. L188-189: It seems biomass consumption 12-25 EJ is not that much compared to your constraint even for the biomass constraint scenario (100 EJ). What is the reason not to produce more bio-based chemicals?
10. L189: Why is CO₂ less relevant as a feedstock? Is it due to production price or constraint in electricity? How much electricity is available for the chemical industry in your scenarios? Some routes would require a lot of electricity and this maybe a constraint factor.
11. L194: Bio-based routes require more mass input to product the same material, and this is partially due to biomass has less carbon content than fossil fuels, as it contains also oxygen. So instead of plotting total mass input, maybe consider plotting the mass of carbon inputs embedded in the feedstock (also apply to Fig 2).
12. L197: What is the limiting factor that fossil fuels remain relevant? Is it the price? Or feedstock constraint (e.g. all biomass is used up? Electricity is used up?)
13. L272 “Without any constraints on biomass use, CCS scale-up, and considering that biomaterials store biogenic carbon, the chemical sector reaches the levels of -746 MtCO₂.yr⁻¹ by 2050 (scenario 1.5C).”: I think this information can be misleading. It sounds as if the chemical industry can reach negative emissions. But most part of the stored carbon will be released within a year and the impact of this emission should be considered. Credits should only be given to durable products with long lifetime that allows storing biogenic carbon for longer time.
14. L283: It is not described how the emissions are calculated. Does it include feedstock acquisition? (e.g. coal production, or agricultural activities for biomass). Also, why is there MNET credit for the MNET_{off} scenario in the figure? Are emissions from the electricity production included (which I think should be)?
15. L369-371: How did the authors come to the assumption of 15%? Why not higher in a future with more demand for oil as feedstock rather than for energy uses?
16. L372: For processes where CO₂ is captured (e.g. biomass gasification), is it considered that the captured CO₂ can also be used in the CCU routes (CO₂ hydrogenation)? This could be relevant for the gCCS scenario.
17. L404-405: Something wrong here.
18. L444: As discussed above, I would suggest having MNET_{off} as the default scenario. The baseline assumption “biomaterials act like a carbon sink provided they are not subjected to incineration or degradation” is not wrong, but most carbon-containing materials (e.g. plastics) have very short lifetime and the carbon will be released. Thus, they are not carbon sink. Also, many countries ban landfilling plastics and I doubt if it is fair to consider it as the default fate for biomaterials.
19. L435: What is the minimum deployment of CCS and how does it compare with your assumption of 10 Gt/year? I think 10 Gt/year CCS is still quite high, especially compared with the 45 Mt/year level we are at today.

20. L442: Does this 100EJ/year limit set for all years? I think this is also too high for the sensitivity scenario. Is it possible to set the limit at the lower bound of IAM scenarios? Also, what are the considered biomass sources? What are the other biomass consumers besides chemical sector?

Reviewer #2 (Remarks to the Author):

This paper assessed the refinery sector with an explicit representation of basic chemicals under climate change mitigation scenarios by using an Integrated assessment model named COFFEE. I think this paper is premature, and its quality is insufficient to be published as a paper in Nature Communications. While I believe the attempt would be worthwhile, the research objective is unclear; easy mistakes can be

seen, and the paper just shows primary model outputs rather than elaborated analysis.

The major comments are as follows:

1) The research objective needs to be clarified. This is really critical because the reviewer cannot assess whether the conclusion is appropriate or not. This would lead to further following essential problems.

2) The abstract and conclusion can mainly be derived without having a model simulation. For example, the following statement taken from the abstract is obvious. What are the other alternatives?

“achieving more than 50% of oil use reduction in 2050 requires a comprehensive feedstocks and material substitution strategy on top of electrification and fuel switch to offset ceased output from global oil refining processes.”

Similarly, in the other sentence of the abstract, I have no alternative idea of how to phase out oil, which is obvious.

“we estimate that without materials efficiency measures and timely scale-up of alternative feedstocks, oil phase-out is unfeasible”

3) There is no description of DAC and hydrogen-related things, which look like critical technology. It is because alternative hydro-carbon (biofuel or synthetic fuel is the only way to phase out fossil oil, and that means carbon source technology and hydrogen are key. Moreover, hydrogen potential and cost are relevant to the variability of renewable energy power generation and its adjustment role.

The hydrogen cost would directly change the condition of synthetic fuel cost, which is highly relevant to the electricity dispatch. However, the current model seems to be unaddressed that part.

4) The overall scenario experimental design is not thoroughly considered. As it has already been pointed out above, the research objective is unclear, and I am not sure how to judge the appropriateness of the scenarios, but let me suppose that the authors wanted to question how much alternative material is needed to achieve a 1.5-degree target for the chemical sector. Then, obviously, the current scenario cannot be answered because CDRs can offset it anyway without changing oil-based petrochemistry under some conditions but such kind of things are not addressed. This is just an example, but either way, without an essential research question, there is no way to judge.

5) The techno-economic assumptions in the supplementary information are impossible to validate. I have gone through the supplementary tables, which show key parameters and source references. I also went through some of the references, but I could not easily figure out how and from what information they are assumed (e.g. technology CR Bionaphtha)

Other minor

1) Why does ammonia need BECCS? Presumably, ammonia does not require carbon and does not need to be BECCS. As described, hydrogen is the essential source, which I intuitively think is available from electrolysis.

2) There is no distinction in the CO₂ sources between fossil combustion, biomass combustion or DAC. How is it modeled? CO₂ can be taken from fossil capture, biomass capture, or atmosphere.

3) The hyperlinks to figures are not working.

Reviewer #3 (Remarks to the Author):

To the authors

Summary

- This is an interesting paper, probably the first I have seen using an IAM, that considers chemical products in a netzero context in detail.

- There are numerous English grammatic errors.
- The referencing system is broken, and I was not able to check claims against the references.
- The paper is interesting, but feels underdeveloped.
- Summary: Major revision.

Editorial

- Minor English issues, e.g., “present an interdependent material and energy systems dynamics” in the abstract should be “present an interdependent material and energy systems dynamic” or “present interdependent material and energy systems dynamics”. I won’t read for English or grammar issues further, but the authors should do a thorough review.

Substantive

- Page 3 lines 61-70. Careful about being overdescriptive of the feedstocks for chemicals. In the main China uses coal and some naphtha for most feedstocks, Europe uses mainly crude oil derived naphtha, and North America C5+ liquids from natural gas, and gas directly. The relative feedstock costs vary.
- Page 5-65, line 16-149 This sentence is confusing “gasoline and naphtha are hydrocarbons that share a similar carbon range, passenger vehicle electrification leads to the use of gasoline streams to meet feedstocks increasing demands in the basic chemicals sector”. Do you mean “increasing feedstock demands in the basic chemicals sector”, ie gasoline replaces naphtha?
- Line 207 “Error reference not found” other after it
- Line 222 Explain more carefully “This results in a higher share of on-purpose routes deployment (Error! Reference source not found.b) to bridge the propylene gap caused by a product split more selective to ethylene than other HVCs in ethane compared to Naphtha S”

- Line 242 Transport electrification maxes out at 37%? This seems on the face of it very low, please explain. Total transport demand for fossil then liquid fuels is critical to your results.
- Line 263 You only get negative emissions with biomass sources for hydrogen if permanent CCS is applied to the carbon, or the carbon is locked permanently in a product (Tanzer & Ramírez, 2019)
- Line 272 The degree of net neutrality of biomaterial storage of carbon is conditional on its longevity, e.g., most plastics are burnt at end of life, which would make them atmospheric neutral depending on the biomass species, how it was harvested, and what happened with the soil carbon as result of harvesting. My understanding is this is not as much of an issue in Brazil, where there is a lots of agricultural waste, but it can be big problem in temperate climates where there are with forest tradeoffs and soil carbon loss. (Hepburn et al., 2019)
- Lines 260-276 Please elaborate more carefully under which conditions and from what processes the chemicals sector and specifically hydrogen production becomes net negative.
- Line 312-325 These are very interesting results that should be drawn out. Also, there are lots of English errors, see above.
- Line 464 The first caveat on not including demand and end of life should come much earlier in the paper. It is critical.
- While I wasn't able to check as I went through because of the "error references..." etc, I am surprised there is no referencing to the IPCC AR6 Industry chapter (Bashmakov et al., 2022)/, specifically sections 11.3 and 11.4. It discussed decarbonization pathways for chemicals in some detail based on the bottom up literature for the sector.

Despite the grammatic issues, thank you for an easy to read & interesting paper, and I hope the journal continues with it.

References

Bashmakov, I., Nilsson, L., Acquaye, A., Bataille, C., Cullen, M., de la Rue du Can, S., Fishedick, M., Geng, Y., Tanaka, K., Bauer, F., Hasanbeigi, A., Levi, P., Myshak, A., Perczyk, D., Philibert, C., & Samadi, S. (2022). Chapter 11: Industry. IPCC AR6 WGIII Mitigation. IPCC.
<https://www.ipcc.ch/report/ar6/wg3/>

Hepburn, C., Adlen, E., Beddington, J., Carter, E. A., Fuss, S., Mac Dowell, N., Minx, J. C., Smith, P., & Williams, C. K. (2019). The technological and economic prospects for CO₂ utilization and removal. *Nature*, 575(7781), 87–97. <https://doi.org/10.1038/s41586-019-1681-6>

Tanzer, S. E., & Ramírez, A. (2019). When are negative emissions negative emissions? *Energy and Environmental Science*, 12(4), 1210–1218. <https://doi.org/10.1039/c8ee03338b>

Point-by-point response to reviewers

Dear Editor and Reviewers,

We are grateful for the thorough review of our manuscript. The thoughtful comments have helped enhance its quality and depth. In this letter, we respond to each point raised, clearly indicating the amendments made to our updated manuscript *Unaddressed non-energy use in the chemical industry can undermine fossil fuels phase-out*.

For the sake of clarity:

- i) Reviewers' comments are written in plain text;
- ii) Our responses are in blue;
- iii) Text quoted from the paper is in *green italics*.

We apologise for the errors in hyperlinks and references in the first version of this manuscript, which have been cleared now. Furthermore, to comply with the journal editorial policies, we changed the title to a version that does not contain punctuation, among other minor formatting changes. The submission also contains a data file integrated to *figshare*, with the results generated by the COFFEE model in the IAMC format, typically used in the IIASA Scenario Explorer and in Model Intercomparison Projects.

We believe these revisions have strengthened the manuscript, and we hope the clarifications we provided below meet the approval.

Reviewer #1:

This manuscript delves into the intriguing topic of the hurdles associated with transitioning away from fossil fuels as the primary feedstock for chemical production. With detailed models of the complex refinery and base chemical production system, this study enriches existing Integrated Assessment Models (IAMs) by incorporating also the chemical sector. However, there are a number of technical issues in the manuscript, particularly with regard to the assumptions behind, remaining to be addressed or clarified, along with some other grammatical and formatting issues. Hence, before I could recommend it for publication, I urge the authors to address the following aspects:

We thank Reviewer #1 for the time taken to review our paper. We genuinely appreciate the balanced and thorough review and thoughtful suggestions. We acknowledge the concerns raised regarding technical assumptions and grammatical and formatting issues. In response, we have carefully revised the entire paper to address these issues comprehensively. Below, we respond to specific comments individually, detailing our actions for each and highlighting significant changes we believe significantly improve the manuscript. This feedback has been invaluable in guiding these revisions.

General comments

Comment 1	In the manuscript, the terms “chemicals” and “materials” have been used interchangeably, while in practice, many don't view it in this way. I would suggest clearly separating them.
Response	We appreciate the reviewer's recommendation. We acknowledge the need for clarity in our terminology to avoid confusion. The general point we wanted to make by using the word “materials” is to stress – particularly to the integrated assessment and energy systems

	modelling communities – that service provision by fossil fuels can happen both as flows (energy via combustion) and as stocks (materials, i.e., plastics, fertilisers, lubricants, asphalt). We want to emphasize this because the “transition away from fossil fuels in the energy system” should also include the transition away from fossil fuels in non-energy uses or in “material systems” in opposition to “energy systems”. Moreover, given that we follow IEA’s definitions to calibrate energy use across sectors, we also need to distinguish between chemicals (i.e., includes both organic and inorganic chemicals) and primary chemicals (HVCs, ammonia and methanol). IEA defines the " Chemical and petrochemical" sector as "[ISIC Rev. 4 Divisions 20 and 21] Excluding petrochemical feedstocks.” which refers to the “Manufacture of chemicals and chemical products” and the “Manufacture of pharmaceuticals, medicinal chemical and botanical products”, respectively (see the ISIC Rev.4, page 48). Thus, the use of the term chemicals in the text will often refer to the Chemical sector (i.e., both organic and inorganic). Therefore, we agree that there are various concepts and definitions that need to be standardized or at least described somewhere in the manuscript or SI. Section S1.2 in the SI attempted to do that, but this comment brought our attention to the need to improve the definition and standardisation of these terms.
Notes/actions	We refined and harmonised these terms in our manuscript and the SI. Section S1.2 of the Supplementary Information clearly outlined specific definitions and distinctions to enhance understanding and ensure consistency. “Materials” is now used only in the introduction to highlight the energy x materials distinction in fossil fuels use. Further, we use “chemicals” or “primary chemicals” whenever suitable in the text. In the revised version of the manuscript we adopted 1) “primary chemicals” to refer to HVCs, ammonia, and methanol; 2) “chemicals” as a more generic term; and 3) “chemicals sector” to refer to inorganic and organic chemicals as defined by the IEA.

Detailed comments

Comment 1	1. L57-58: Perhaps reformulate the statement to make it clearer that it’s the primary chemicals production contributing to ca. 1 Gt CO ₂ /yr.
Response	We thank the reviewer for pointing this out. We agree that the way we have formulated the sentence on lines 57-58 could lead to misunderstanding regarding to what the verb “contributing” was referring to.
Notes/actions	To address this concern we reformulated the statement as: “Of this, two-thirds are used as feedstocks for primary chemicals production (IEA, 2022, 2018), which contributes to approximately 1 GtCO₂yr⁻¹ of emissions”.
Comment 2	2. L73-74: Synthetic polymers, fertilizers, lubricants, asphalt and solvents, etc. are all chemicals. Please revise the sentence here.
Response	Thank you for highlighting this. We recognize the need for further clarity and harmonization in the terms we use along the article. As mentioned in Comment 1 (General comments), we want to stress the distinction between energy and material use regarding fossil fuels.

Notes/actions	We refined and harmonized terms such as chemicals , primary chemicals and materials in our manuscript and SI.
--

Comment 3	3. L120-122: I don't fully get it here, how come the scenario with all the restrictions has the lowest CO2 emissions (i.e., overall we can do better without CCS and biomass as feedstocks for the chemical industry?) and how the authors come to the conclusion "that CCS and biomass availability have a critical role in mitigation across all sectors" (i.e., if the previous statement is true, then why CCS and biomass availability is important?). It would be good to provide some elaboration here.
------------------	--

Response	We thank the reviewer for urging us to clarify these results. These constraints affect not only the chemical sector but also the whole IAM. Therefore, the 100 EJ/yr limit to biomass affects bio-based HVCs production but also Fischer-Tropsch Biomass-to-Liquids processes, conventional biofuels production and all BECCS technologies, for example. Actually, as a linear optimization model, COFFEE minimizes its cost function subject to a set of constrains. In our case, these constraints include CO₂ emissions targets for various scenarios, such as the 1.5°C target with additional restrictions on biomass use (PBIO), Carbon Capture and Storage (CCS) (gCCS), and the exclusion of biogenic carbon storage (MNEToff), as well as a scenario combining all these limitations (all). Introducing more stringent constraints, the model is forced to find alternative pathways to achieve the emission reduction targets. Particularly, limiting the use of biomass and CCS reduces the model's flexibility in achieving climate targets (via BECCS), especially in sectors that are hard to decarbonize or difficult to electrify. This search for alternatives leads to the deployment of more costly mitigation measures. Given that our model operates over the 2010-2100 period and applies an annual discount rate of 5%, the cost of future mitigation efforts is adjusted. Thus, by 2030, scenarios with all restrictions show the lowest CO₂ emissions. This "early action" allows for a cost reduction later since the options available are more expensive. However, by 2040, this paradox resolves, with these more restricted scenarios exhibiting higher annual emissions levels than others. Therefore, our objective with the referred sentence was to highlight that the stricter the constraints, the more immediate, rapid and extensive our climate action must be. Therefore, the issue is not that emissions are lower when there is less CCS and biomass; rather, it means we must accelerate our efforts to achieve the same temperature goals as would be possible without these limitations. We agree with the reviewer that our presentation about it might lead to misinterpretation of our results, which shows that, in more stringent scenarios, the earlier the climate action needs to happen and in a more challenging scale and speed. As described below, we hope that our amendment adequately addresses the concern raised.
-----------------	--

Notes/actions	In light of your feedback, we have rephrased the whole paragraph: FROM: "Our results show that, the more restrictions on biomass, CCS, and biogenic carbon storage (or material net – MNET) are considered, the steeper are the CO2 emissions pathways to achieve 1.5oC goals,
--

	particularly in 2030 (Figure 1). Global CO₂ emissions in the 1.5C and MNEToff scenarios are reduced from 36 GtCO₂yr⁻¹ in 2010 to approximately 18 GtCO₂yr⁻¹ in 2030 globally, whereas the scenario with all restrictions reaches around 12 GtCO₂yr⁻¹ that same year. This illustrates that CCS and biomass availability have a critical role in mitigation across all sectors. These restrictions also explain the difference between achieving global net-zero CO₂ emissions around 2050 or 2060.” TO: “Our results show that scenarios with stricter restrictions on global biomass availability, deployment of CCS, and biogenic carbon storage (or material net – MNET) require taking climate action faster, sooner, and on a larger scale, as demonstrated in Figure 1-a. Global CO₂ emissions in the scenario with all restrictions must be reduced by at least 6 GtCO₂ per year less than in scenario 1.5C by 2030 to stay within the CO₂ budget. This illustrates that CCS and biomass availability are critical in mitigation across all sectors, including but not limited to the primary chemicals. These restrictions also explain the difference between achieving global net-zero CO₂ emissions around 2050 or 2060.”
--	---

Comment 4	4. Please replace Figure 1 with a high-resolution version and increase the font size to make it readable.
Response	We thank the reviewer for pointing out the need for enhanced readability in Figure 1. We would like to clarify that all figures in the manuscript, including Figure 1, were generated at a resolution of 300 dpi to ensure high quality, in accordance with the submission system's requirements. Additionally, we have adhered to the font size and figure formatting guidelines as outlined by Nature. Despite these measures, the automatic conversion to PDF might have automatically reduced the quality of the figure. We will be more careful to ensure that the quality of the figures is not reduced.
Notes/actions	N/A

Comment 5	5. L131-132: This seems to be contradictory to L106-107.
Response	We thank the reviewer for bringing attention to this; they are correct. We apologize for the oversight during one of our internal reviews; we have now corrected this mistake.
Notes/actions	We corrected the sentence in L106-107 based on the reviewer's recommendation to: “(i) the assumption that biogenic carbon is stored in unrecycled/unrecovered biomaterials is turned off (MNEToff)”

Comment 6	6. L138-139: I think it would be good to add some elaboration on the causes here, esp. the increased use of coal.
Response	We thank the reviewer for this suggestion. The resurgence of coal use of 4-35 EJ, identified in COFFEE after mid-century, aligns with results observed by Achakulwisut et al., (2023) across certain model families while analysing C1, C2, and C3 scenarios from the AR6 scenario database. The factors for coal resurgence in COFFEE are outlined as follows:  The challenge is not only to reduce emissions and fossil fuels use in current demand levels; it also involves meeting the growing demand for energy services in regions expected to see a surge in

	demand after mid-century. We observe that coal use over 2 EJ/yr happens in Africa, Southeast Asia, and India regions. These are the regions projected to increase population and affluence after 2050, thereby stressing the energy supply module.  • Oil and gas costs become more expensive after mid-century, which also contributes to this effect. • Coal use nearly ceases until 2050, particularly in applications without carbon capture and storage technology (CCS). The resurgence after 2050 happens with CCS and only in a select regions and sectors. In this cases, coal is difficult to substitute, for instance as a reducing agent in steel production.
Notes/actions	We added a sentence to this paragraph to elaborate on that: The resurgence in coal use observed after mid-century aligns with the literature and is attributed to the combining effects of: 1) a rising energy demand in regions projected to experience significant population and affluence growth post-2050 (i.e., Africa, Southeast Asia, and India), 2) escalating costs of oil and gas, and 3) coal use transitioning to include carbon capture³.

Comment 7	7. L164-165: How so? Any reference for this statement?
Response	We thank the reviewer for the interest in this matter, which is relevant for our analysis. This statement is grounded in the fundamental principles of petroleum refining technology and economics. Due to the capital-intensive nature of oil refineries, there are inherent limitations in modifying the existing refinery infrastructure. It is possible to adjust operation parameters within certain bounds to optimise product outputs based on market demands. For instance, conventional Fluidized Catalytic Cracking (FCC) units, which convert heavy oil fractions into lighter streams with higher value, present a propylene yield of roughly 2-5 wt%. The propylene yield in FCC units can be increased up to 15% by modifying the residence time, temperature of reaction, and the catalyst-to-oil ratio, for instance (see refs below). This would result in an increased petrochemical integration, which could also entail some small retrofits in specific units. However, increasing the overall petrochemical yield of a fuel-oriented refinery would require a complete reconfiguration, from shutting down units that are no longer needed to installing new units or even completely new plants. Given the long-lived nature of refinery equipment, this is usually prohibitively costly. Moreover, the current market conditions favours fuel production over non-energy petrochemical products, adding to the financial challenge. Nevertheless, we re-read the whole paragraph and came to the conclusion that we are overemphasizing refineries' limited flexibility. Thus, besides including the references that support their limited flexibility, we also deleted the last sentence to avoid confusion. References: John, Y.M., Patel, R. & Mujtaba, I.M. Maximization of propylene in an industrial FCC unit. Appl Petrochem Res 8, 79–95 (2018). https://doi.org/10.1007/s13203-018-0201-1 Akah, A., Al-Ghrami, M. Maximizing propylene production via FCC technology. Appl Petrochem Res 5, 377–392 (2015). https://doi.org/10.1007/s13203-015-0104-3

Notes/actions	To support our statement about refineries' limited flexibility, we added the following references in the first sentence and deleted the last sentence of the referred paragraph, as below: Refineries are capital-intensive plants with limited flexibility concerning their product outputs^{23,24}. They are designed to optimize profit margins, which are influenced by the quality of crude oil and the product slate required by consumers. Typically, the complexity of refinery operations increases when the crude input shifts from sweet and light to sour and heavy grades, which requires advanced processing units such as Hydrocracking (HCC). The product slate is primarily focused on gasoline and diesel but may also extend the production of specialized products like aviation fuels, petrochemical naphtha, and lubricants, depending on market demands. Ref23: J Bengtsson, D Bredström, P Flisberg & M Rönnqvist (2013) Robust planning of blending activities at refineries, Journal of the Operational Research Society, 64:6, 848-863, DOI: 10.1057/jors.2012.86 Ref24: de Barros, M.M., Szklo, A., 2015. Petroleum refining flexibility and cost to address the risk of ethanol supply disruptions: The case of Brazil. Renewable Energy 77, 20–31. https://doi.org/10.1016/j.renene.2014.11.081
--

Comment 8	8. L173: Is the increase in capacity utilisation due to the phase out of old capacity? What does this increase mean for the chemical industry?
Response	We thank the reviewer for this question. The COFFEE model, as a perfect-foresight, intertemporal linear optimization model, is designed to minimize costs throughout the entire period of analysis. By identifying cost-optimal solutions under specific constraints, COFFEE strategically selects investment paths that prevent stranded assets, thereby maximizing the value and performance of new capital over its entire life-cycle. In this sense, the logic of the model is to shrink oil refining capacity to the least required to fulfil demand. Therefore, while “old” capacity reduces its utilisation factor until its complete phase-out, the new capacity is right-sized for demand and operates at the maximum utilisation factor. Within this framework, the chemical sector, which predominantly relies on fossil-based feedstocks and requires refined products, plays a critical role in shaping the model's decisions on new processing capacities. These decisions include the selection of regions for new facilities, the quality of oil produced, and the choice of oil fields. The sector affects the model's decision, for instance, when feedstock substitution is either uneconomical or constrained by the unavailability of alternative feedstock. This results in refining capacity to remain operative or to drive the construction of new refineries with increased HVCs output.
Notes/actions	To make the abovementioned dynamics of capital optimization, we made changes in the referred paragraph as follows: FROM: “Our results show three trends in the refining sector in mitigation scenarios, which intensify in the more restrictive scenarios. Firstly, the

	refining sector overall shrinks in terms of capacity (see Figure S8 in the SI) throughout the century and also its utilisation factor until 2050. Total refining capacity in terms of oil input reduces from ~191 EJ.yr-1 in 2010 to 103 (1.5C) -132 (all) EJ.yr-1 in 2050 and 44 (gCCS) – 77 (all) EJ.yr-1 in 2070. The upper limits are set by the scenario all, not NPi, indicating a greater need for oil in more constrained scenarios. Furthermore, the utilisation factors drop from ~70% in 2020 to 30-50% in mitigation scenarios by 2030, and then increase to 100% in 2070 after new capacity is built.” TO: “Our results highlight three trends in the refining sector in mitigation scenarios, which intensify in the more restrictive scenarios. Firstly, the refining sector shrinks in capacity throughout the century (see Figure S10 in the SI); its utilisation factor also reduces until 2050. Total refining capacity in terms of oil input reduces from ~191 EJ.yr-1 in 2010 to 103 (1.5C) -132 (all) EJ.yr-1 in 2050 and 44 (gCCS) – 77 (all) EJ.yr-1 in 2070. The upper limits are set by the scenario all, not NPi, indicating that highly restricted scenarios foresee a greater demand for oil. Furthermore, the utilisation factors drop from ~70% in 2020 to 30-50% in mitigation scenarios by 2030, then increase to 100% in 2070 after new capacity is built. This follows the dynamics of capital optimization to maximize long-term value of newly built capacity considering energy and non-energy demands.
--	---

Comment 9	9. L188-189: It seems biomass consumption 12-25 EJ is not that much compared to your constraint even for the biomass constraint scenario (100 EJ). What is the reason not to produce more bio-based chemicals?
Response	The reviewer raises an important point regarding the apparent discrepancy between the figures of biomass consumption in the chemical sector (12-25 EJ) and the biomass constraint (100 EJ) outlined in our study. The 100EJ biomass constraint applies to the whole integrated assessment model; thus, mitigation measures based on biomass across sectors compete for the limited availability of biomass e.g., raw material for biochemicals, production of advanced biofuels via Fischer-Tropsch synthesis with CCS, not to mention current biomass uses such as in the paper and pulp industries. Therefore, the biomass use of 12-25EJ is the level allocated to the chemical sector when competing for other uses.
Notes/actions	To clarify this issue in the manuscript, we changed the referred sentence as follows: FROM: Results indicate that in all mitigation scenarios, there is an increase in liquid and solid biomass from 6-25 EJ.yr-1 in 2050 to 12-25 EJ.yr-1 in 2070. TO: Results indicate that mitigation scenarios see an increase in liquid and solid biomass use in the chemical sector, reaching 6-25 EJ.yr-1 by 2050 and 12-25 EJ.yr-1 by 2070

Comment 10	10. L189: Why is CO2 less relevant as a feedstock? Is it due to production price or constraint in electricity? How much electricity is available for the
-------------------	--

	chemical industry in your scenarios? Some routes would require a lot of electricity and this maybe a constraint factor.
Response	The reviewer raises a key question about CO₂ relevance as a feedstock. COFFEE does not apply constraints in electricity generation. Thus, the model can deploy conventional or alternative electricity generation technologies to:  1) Fulfil electricity demand in the current level of participation, 2) Electrify further energy services, and 3) Produce hydrogen or store electricity in batteries. The reasons for which CO₂ is less relevant as feedstock in our results are listed below:  1. CCU and CCS compete for the same carbon dioxide sources in COFFEE: There are 2 pathways through which CO₂ as a feedstock is sourced to the chemical sector in COFFEE. 1) From fossil- or bio-based concentrated sources such as power plants, cement kilns, blast furnaces, and methanol/ammonia plants (via gasification/partial oxidation, but not electrolysis). Both energy and process CO₂ emissions can be captured from concentrated sources, and costs for CO₂ transportation are accounted for, although in a stylized and simplified manner. 2) From diluted CO₂ in the atmosphere via Direct Air Capture (DAC). Captured CO₂ thus feeds into a “regional pool” in the model, where the model decides whether it is geologically stored or used for CCU. 2. In very ambitious carbon stringent scenarios, the model decision weights towards negative emissions: not only CCU via DAC or other carbon sources are more expensive than other carbon sources, but they also reduce their participation in highly ambitious scenarios because BECCS or other CDR measures become critical to stay within the carbon budget. Technologies such as FT-BtL can provide energy/material services while removing CO₂ from the atmosphere. In other words, and from an optimization model perspective, it is much more costly to 1. Burn fossil/biofuels to produce CO₂, 2. Capture and transport this CO₂, and 3. Hydrogenate this CO₂ to methanol and further into olefins. In this case, the carbon balance is at most (and most probably not) neutral. Even so, in stringent scenarios, DAC deployment reaches levels of ~700MtCO₂ by 2100 globally, indicating it can be a relevant carbon source where biomass is not immediately available. To sum up, the model is not trying to find the optimal solution for the chemical sector or to provide the pathway that leads to a chemical industry net-zero by 2050. In our case, from a linear-optimization integrated assessment perspective, the model aims for a cost-optimal solution of achieving global temperature goals in the whole system, including the transport, electricity, buildings, industry, and land-use sectors. This leads to competition for resources in those sectors. Therefore, resources are best allocated when they lead to overall emissions reduction. Future studies focusing on how policy incentives could lower the costs of CCU would be valuable. For instance, integrating electrolysis-based hydrogen production with the use of oxygen produced as a by-product in oxy-fuel processes could result in a more concentrated CO₂ stream from

	combustion processes. This concentrated CO ₂ is easier to capture and purify for further utilisation in the chemical sector.
Notes/actions	Based on this and other reviewers' concerns, we added a section in the SI file to describe how carbon sources are modelled in COFFEE. A shorter section was added to the Methods section in the Manuscript.

Comment 11	11. L194: Bio-based routes require more mass input to product the same material, and this is partially due to biomass has less carbon content than fossil fuels, as it contains also oxygen. So instead of plotting total mass input, maybe consider plotting the mass of carbon inputs embedded in the feedstock (also apply to Fig 2).
Response	We thank the reviewer for highlighting this and for the suggestion. We agree that this information complements the energy and total mass data that we have provided, as both bio-based and CO ₂ have lower carbon content compared to fossil fuels. We chose to maintain the graph with total mass as it is relevant for understanding the scale of materials involved and the logistical challenges associated with their transportation and capture (for CO ₂).
Notes/actions	In response, we have included a new figure based on the previous, substituting total mass for carbon mass next to the energy values. We included the new figure in the SI to maintain a concise and focused narrative in the main text. We have also made reference in the main text to this figure in the SI for readers interested in the specifics of carbon mass.

Comment 12	12. L197: What is the limiting factor that fossil fuels remain relevant? Is it the price? Or feedstock constraint (e.g. all biomass is used up? Electricity is used up?)
Response	The reviewer raised a question about the limiting factors that make fossil fuels remain relevant. As mentioned above, there are no constraints for electricity generation and use in the model. The same applies to feedstock (except, of course, for the scenarios that include those constraints and the resource/land availability). COFFEE, as an integrated assessment model, minimizes costs under constraints considering that mitigation measures across sectors compete for resources. Therefore, the least-cost pathway to 1.5 degrees still requires 38-72% of feedstocks to be fossil-based; with more fossil fuels, the more challenging the scenario is. This is due to both restrictions in alternative feedstock availability (both biomass and CO₂, restricted by CCS), but also to economies of scale and scope that oil refineries benefit from. In other words, it is less expensive to keep using oil to some extent and reaching 1.5-degrees targets in the scenarios considered. However, we recognize that the way we phrased the sentence about feedstock substitution left a "negative" impression about the extent to which fossil-based feedstock could be substituted while results show somehow the opposite. Substituting up to 62% of fossil-based feedstocks by 2050 means huge efforts for developing new transportation logistics infrastructure, besides the fast and impressive change in technology portfolio (in less than three decades). Given the age profile of fossil-based capacity recently deployed in Southwest Asia, for instance, this is quite an ambitious goal (https://www.argusmedia.com/en/news-and-insights/latest-

	market-news/1599017-viewpoint-se-asian-petrochemical-investments-surge)
Notes/actions	We rephrased that sentence to emphasize the reduction of fossil fuels use by 2050 while also highlighting the reasons for this. FROM: In any case, fossil fuels remain relevant regardless of the mitigation scenario, although they are reduced by around 28-62%, in 2050, and 25-54% in 2070 respective to NPi results in those years. The upper bound refers to the gCCS scenario, indicating a higher opportunity for BECCS coupled with materials use in long-term applications. The lower bound refers to the all scenario, reflecting the increased need for oil when all restrictions are applied and feedstocks substitution is limited. However, as Figure S7 in the SI shows, this oil production gap is met primarily with light and medium sweet oil. These qualities of crude oil not only lower emissions but also yield a higher output of products per input of raw material²⁷⁻²⁹. TO: Fossil fuels used as feedstocks in mitigation scenarios achieve reductions of up to 62%, in 2050, respective to NPi. This is quite ambitious considering challenges such as developing new supply chains and transportation logistics for alternative feedstocks. At least 500Mt of primary chemicals capacity is less than 20 years old, and massive investments in fossil-based capacity have been made in the past 10 years, particularly in China and Southeast Asia^{29,30}. Significantly, scenarios that present reduced petroleum use as carbon feedstock are compatible with reduced oil production (Figure S8 in SI). The upper bound of feedstock substitution (62%) refers to the gCCS scenario, indicating a higher opportunity for BECCS and materials use in long-term applications. The lower bound (28%) refers to the scenario all, reflecting the increased need for oil when alternative feedstock availability is constrained. However, as Figure S8 shows, reminiscent oil demand is met primarily with light and medium sweet oil. These qualities of crude oil have lower emissions and yield a higher output of products per input of raw material³¹⁻³³.

Comment 13	13. L272 “Without any constraints on biomass use, CCS scale-up, and considering that biomaterials store biogenic carbon, the chemical sector reaches the levels of -746 MtCO₂.yr⁻¹ by 2050 (scenario 1.5C).”: I think this information can be misleading. It sounds as if the chemical industry can reach negative emissions. But most part of the stored carbon will be released within a year and the impact of this emission should be considered. Credits should only be given to durable products with long lifetimes that allows storing biogenic carbon for a longer time.
Response	We thank the reviewer for this remark. This is indeed a critical conclusion of our work. First, we list below a few points to clarify this message:  • The -746 MtCO₂.yr⁻¹ by 2050 is attributed both to material storage and carbon capture of bio- based CO₂ from ammonia production. Material storage alone accounts for -465 MtCO₂.yr⁻¹ by 2050. • We account for emissions according to the End-of-Life of HVCs. We used country-level data aggregated to the regional level based on current levels of incineration, landfilling/mismanagement, and recycling using the World Bank’s ‘What a waste’ database. We assumed that these levels remained constant over the century,

	which we acknowledge as a limitation of the study to be addressed in future studies. Emissions for incineration and recycling are taken into account. Therefore, while Europe, Korea, and Japan have a high incineration rate (and those emissions are accounted for), that is rarely true for other regions. As landfilling is still relevant, all the carbon that has this destination by the end of its lifetime is considered to be stored. This is done for carbon accounting reasons, not to support landfilling. The reason why we have a plastic accumulation and pollution problem is precisely because of its stability and non-degradability. Hence our modelling approach aims to capture that effect as well. This approach was similarly adopted by Stegmann et al. (2022). To clarify, we agree with the reviewer that bioplastics that end up in landfills of are otherwise mismanaged should not be eligible for carbon credits. We only argue that we need to be careful about carbon accounting of plastics throughout their life-cycle. Addressing other sustainability factors in addition to decarbonisation strategies falls outside the scope of our current discussion. Hence, our findings are currently focused on exploring the potential of bio-carbon storage in long-lived applications, which is represented by the share of bioplastics that are neither recycled nor incinerated, i.e., landfilled. In essence, carbon credits should not be awarded to plastics that are simply thrown away, yet it is undeniable that such products do store carbon. If this were not the case, plastic pollution would not be as significant an issue. Carbon accounting is critical for reliable modelling practices. This is one of the reasons why we ran the MNEToff scenario as a sensitivity rather than the baseline. References: Stegmann, P., Daioglou, V., Londo, M. et al. Plastic futures and their CO2 emissions. Nature 612, 272–276 (2022). https://doi.org/10.1038/s41586-022-05422-5
Notes/actions	The referred sentence was rephrased for clarification as follows: FROM: Figure 4 shows the role of the chemical sector in a 1.5°C world subject to different restrictions. Without any constraints on biomass use, CCS scale-up, and considering that biomaterials store biogenic carbon, the chemical sector reaches the levels of -746 MtCO₂.yr⁻¹ by 2050 (scenario 1.5C). Not only the chemical sector does not always behave as hard-to-abate from an emissions standpoint, but it can also be a climate asset. Emissions reduction stem mostly from BECCS in ammonia production and biogenic carbon storage, as well as increased efficiency in the fossil fuels-based platforms. TO: Figure 4 shows the role of the chemical sector in a 1.5°C world subject to different restrictions. It achieves a net reduction of -0.73 GtCO₂.yr⁻¹ in scenario 1.5C and -1GtCO₂.yr⁻¹ in scenario gCCS by 2050. Direct emissions reduction stems mainly from BECCS in ammonia production and biogenic carbon storage, as well as increased efficiency in fossil fuels-based platforms. (We updated the value of net reduction also based on comment 14).

Furthermore, we included a section in the SI file to clarify EoL emissions from plastics production:

S2.4. Final disposal assumptions

Solid waste management data from World Bank's What a Waste 2.0 database was used as a baseline to estimate regional shares for incineration, landfilling (i.e., controlled landfilling plus mismanagement), and recycling (see Figure S4). Incineration and recycling emissions were calculated based on ref.82. For plastics incineration, it was assumed that the carbon content of each HVC was emitted as CO₂. Therefore, 3.14 tCO₂/t ethylene, 3.14 tCO₂/t propylene, 3.25 tCO₂/t butadiene, and 3.38 tCO₂/t aromatics (using benzene as a proxy) were assumed when accounting for those emissions. Recycling considers 10% of material loss at every cycle and emissions account for electricity use from shredding, extrusion, and agglomeration.

Figure S4. Final disposal shares assumed in the base year for COFFEE regions. Source: own elaboration based on ref. ⁸³.

We also added a sentence in the Results section/Emission pathways for the global chemical sector:

While accounting for biogenic carbon storage in bioplastics highlights the potential of carbon removal in long-lived application as a mitigation strategy, it is essential to assess other sustainability dimensions to evaluate its effectiveness as a carbon mitigation strategy fully.

Comment 14

14. L283: It is not described how the emissions are calculated. Does it include feedstock acquisition? (e.g. coal production, or agricultural activities for biomass). Also, why is there MNET credit for the MNEToff scenario in the figure? Are emissions from the electricity production included (which I think should be)?

Response

We thank the reviewer for the question about emissions accounting and the remark on MNET credit.

On the emissions, we calculate gate-to-gate emissions of the chemicals sector, which includes the explicitly modelled products (HVCs, ammonia, and methanol) and the implicitly modelled ones (other chemicals, as described in the glossary section in the SI). This means emissions resulting from combustion within the boundaries of chemical plants resulting from combustion processes and chemical reactions (i.e., energy and process emissions, respectively). Indirect emissions arising from

	electricity production are not allocated to the chemical sector. Instead, the electricity use within chemical plants is accounted for as and used as an input to the electricity module. This module comprehensively evaluates all electricity demands – including illumination, electric vehicle charging, which takes into account along with all the other electricity demands such as illumination, electric vehicles, space/water heating, air conditioning, refrigerators, among others. These aggregated demands are considered in the overall optimization for capacity deployment and utilisation factor. Consequently, the module calculates the associated emissions, which are reported as “electricity sector emissions”. The same is valid for emissions of feedstock acquisition, which take place in the resource extraction (petroleum/coal) and land-use modules (biomass). In sum, all emissions associated with the energy system (from extraction, refining, transport to use) are accounted for by the integrated assessment model. On the MNET credit in the scenarios MNEToff and all, we thank the reviewer for bringing our attention to this. This was a typo in our code – MNET credits are given also to other materials, but we only turned off MNET for plastics. The remaining MNET are thus referring to carbon storage in other materials.
Notes/actions	We changed the following sentences in the text as follows: FROM: Emissions reduction stem mostly from BECCS in ammonia production and biogenic carbon storage, as well as increased efficiency in the fossil fuels-based platforms. TO: Direct emissions reduction stem mainly from BECCS in ammonia production and biogenic carbon storage, as well as increased efficiency in the fossil fuels-based platforms. FROM: Global CO₂ emissions from the chemical sector. Includes direct emissions from basic chemicals (explicitly modelled in COFFEE) and other chemicals (implicitly modelled in COFFEE) (see the definitions section in the SI). TO: Global direct CO₂ emissions from the chemical sector. Includes direct (Scope 1) emissions from basic chemicals (explicitly modelled in COFFEE) and other chemicals (implicitly modelled in COFFEE) (see the definitions section in the SI). Furthermore, the figure and net emissions were updated in the manuscript to account for MNET credits related to plastics only. FROM: It achieves a net reduction of -746 MtCO₂.yr-1 by 2050 in scenario 1.5C. TO: It achieves a net reduction of -0.73 GtCO₂.yr-1 by 2050 in scenario 1.5C.

Comment 15	15. L369-371: How did the authors come to the assumption of 15%? Why not higher in a future with more demand for oil as feedstock rather than for energy uses?
Response	We thank the reviewer for this question. COFFEE currently represents refineries focused on producing fuels. In COFFEE, these technologies are equipped with the current level of integration to the petrochemical sector

	i.e., propylene and BTX co-production, which is represented at the regional level, as average propylene and BTX co-production per refinery capacity per region. The majority of crude oil is processed to produce gasoline, diesel, and in lesser extent also aviation and maritime fuels. From a refinery's standpoint, the production of propylene and BTX is currently minor in scale and value when compared to fuels production. However, it holds significant importance for the petrochemical industry. As elaborated above in Comment 7, fuel-oriented refineries have limited flexibility to adjust their operation parameters to enhance propylene output in FCC units without changing their configuration. That limited flexibility is what we aimed at representing here. This capability to slightly increase propylene output—by up to 15%, as noted by Akah and Al-Ghrami (2015)—serves as a benchmark for what is known as high-severity operations in refinery units, which we also assumed to BTX co-production in Catalytic Reforming units. Looking ahead, we agree that there is an anticipation that oil will predominantly be used as feedstock rather than for energy purposes. To accurately depict this shift, it would be necessary to conceptualize refineries that process crude oil into exclusively non-energy products, which have been termed as crude oil to chemicals (COTC) refineries. These would require comprehensive data on techno economic parameters and detailed process engineering analysis that accurately represent units and utilities, which is currently lacking in the literature or in industry reports. Unfortunately, this is a limitation in our current research, which we aim to address by exploring this emerging area in future studies. References: Akah, A., Al-Ghrami, M. Maximizing propylene production via FCC technology. Appl Petrochem Res 5, 377–392 (2015). https://doi.org/10.1007/s13203-015-0104-3
Notes/actions	Based on this comment, we changed the “Oil production and refining sector” subsection in the Methods section to clarify what type of refineries we are representing. FROM: COFFEE’s resolution of the oil and gas sector is in general higher than in other IAMs, which includes the representation of oil qualities, crude trade, and refinery typologies, TO: COFFEE’s resolution of the oil and gas sector is generally higher than that present in other IAMs, which includes the representation of oil qualities, crude trade, and fuel-oriented refinery typologies, We also included the missing reference in the Methods section to justify the 15% increase output: FROM: We also assumed that greenfield refinery capacity could increase its HVC output up to 15% to assess potential opportunities of increased petrochemical integration. TO: We also assumed that greenfield fuel-oriented refinery capacity could increase its HVC output up to 15% to assess potential opportunities for increased petrochemical integration under high-severity operation, based on ref.³³ .

Comment 16	16. L372: For processes where CO ₂ is captured (e.g. biomass gasification), is it considered that the captured CO ₂ can also be used in the CCU routes (CO ₂ hydrogenation)? This could be relevant for the gCCS scenario.
Response	We thank the reviewer for your question. Yes, the reviewer is correct. As mentioned in the response to Comment 10 , CCU and CCS compete for the same carbon dioxide sources in COFFEE. Therefore, the rationale behind developing the gCCS scenario was to 1) limit CO ₂ as a feedstock, 2) identify model prioritization of CCS or CCU in carbon captured restricted realities, and 3) understand the extent to which biomass use is relevant for mitigation scenarios if CCS is not available.
Notes/actions	We rephrased a sentence in the Main section to clarify that the scenario design aimed at limiting feedstock production while keeping temperature increases up to 1.5-degrees as follows: FROM: We also test a set of restrictions in different pathways aligning with a below 1.5o C scenario, namely: (i) the assumption that biogenic carbon is stored in unrecycled/unrecovered biomaterials (MNEToff); (ii) limited scale-up of global carbon capture and storage (gCCS), which also affects the availability of CO₂ as carbon feedstock; (iii) limited global biomass availability (PBIO), which affects both energy and non-energy applications, and (iv) a combination of all aforementioned restrictions (all). TO: We also test a set of restrictions to limit alternative feedstock availability throughout the century in different pathways aligning with a below 1.5oC scenario, namely: (i) the assumption that biogenic carbon is stored in unrecycled/unrecovered biomaterials is turned off (MNEToff); (ii) limited scale-up of global carbon capture and storage (gCCS), which also affects the availability of CO₂ as carbon feedstock; (iii) limited global biomass availability (PBIO), which affects both energy and non-energy applications, and (iv) a combination of all restrictions above (all). We also added a section in the Online Methods and Extended Methods sections on the sources of carbon-based feedstock. In Online Methods: Sources of carbon-based feedstock Carbon-based feedstocks included in COFFEE are categorized as fossil-based, bio-based, and CCU-based, representing products derived from CO₂ conversion. Fossil-based feedstocks include coal, natural gas, and products derived from crude oil processing in refineries, including LPG, naphtha, and heavy oil. Notably, gasoline can be converted to naphtha and vice-versa. Bio-based feedstocks include bioethanol and co-products of Fischer-Tropsch Biomass-to-Liquids synthesis (FT-BtL) such as bio-naphtha and bio-LPG. These bio-based feedstocks derive from the processing of agricultural raw materials including sugary, starchy, lignocellulosic and oily crops. Currently, the only CCU-based feedstock represented is methanol, when produced via the hydrogenation of CO₂ captured from diluted (i.e. atmosphere via Direct Air Capture based on CO₂ absorption in sodium hydroxide solution) or concentrated sources. Concentrated sources include blast furnaces, power plants, cement kilns, and bio- or fossil-based hydrogen production processes, for example. CO₂ captured feeds into a regional "CO₂ pool" that can be either geologically stored or utilised as feedstock. Section 2.1 in the SI presents a more detailed description of the abovementioned feedstocks.

	In Extended Methods: COFFEE represents fossil-, bio- and CCU-based feedstocks as options for producing primary chemicals. Fossil-based feedstock: Primarily includes natural gas (i.e., methane and ethane), oil (i.e., LPG, naphtha and heavy oil), and coal. Natural gas can be produced in fields associated with oil or in dedicated fields. Ethane and methane can be recovered from natural gas and used for steam cracking and steam methane reforming units, respectively. LPG, naphtha and heavy oil are outputs of the petroleum refining sector, which is described in ref.¹². LPG is an input to propane dehydrogenation process, naphtha can be used in steam cracking and catalytic reforming units whereas heavy oil is used for hydrogen to ammonia production via partial oil oxidation. Coal is produced in different qualities in the resource module (bituminous, subbituminous, and lignite) but only bituminous coal is allowed in coal gasification processes. Bio-based feedstock: Primarily includes solid and liquid biofuel produced from planted forests and energy crops. Solid biofuel is obtained from woody (eucalyptus) and grassy sources as well as agriculture residues, including sugarcane bagasse. Eucalyptus can be used for biomass gasification processes to produce hydrogen (for energy or ammonia production), methanol, or can be used in Fischer-Tropsch Biomass-to-Liquids (FT-BtL) to produce advanced liquid biofuels. Sugarcane bagasse can be used for 2G bioethanol production. Liquid biofuels included as feedstocks for the chemical sector are bioethanol (via fermentation/distillation from sugarcane, maize, wheat, sugar beet, and agricultural residues), and bio-naphtha, and bio-LPG as co-products of hydrotreated vegetable oils (HVO) or hydroprocessed esters and fatty acids (HEFA) production from soybean oil, maize oil, or animal fat. HVO and HEFA are advanced biofuels alternatives to be used in diesel engines as green diesel or jet fuel (or sustainable aviation fuel – SAF), and might co-produce bio-naphtha and bio-LPG to be used in other sectors, including as feedstocks. Biomethane could be a promising alternative feedstock. However, its supply chain is not yet developed in COFFEE and it remains as a topic to be further explored in future work. CCU-based feedstock: CO₂ as a feedstock is sourced from a “CO₂ pool”, which is composed by CO₂ captured from: 1) fossil- or bio-based concentrated sources such as power plants, cement kilns, blast furnaces, and methanol/ammonia plants (via gasification/partial oxidation, but not electrolysis). Both energy and process emissions can be captured from concentrated sources, and costs for CO₂ transportation are considered; 2) diluted CO₂ in the atmosphere via Direct Air Capture (DAC) based on absorption in sodium hydroxide solution, based on refs.^{13,14}. Captured CO₂ feeds into a regional “CO₂ pool” in the model, where the model decides whether it is geologically stored or used for CCU.
--	---

Comment 17	17. L404-405: Something wrong here.
Response	We apologize for the formatting issues. These errors likely occurred during the automatic conversion to PDF format within the online submission system.
Notes/actions	To avoid these issues, we now pre-generate the pdf prior to submission.

Comment 18	18. L444: As discussed above, I would suggest having MNEToff as the default scenario. The baseline assumption “biomaterials act like a carbon sink provided they are not subjected to incineration or degradation” is not wrong, but most carbon-containing materials (e.g. plastics) have very short lifetime and the carbon will be released. Thus, they are not carbon sink. Also, many countries ban landfilling plastics and I doubt if it is fair to consider it as the default fate for biomaterials.
Response	We thank the reviewer for this thoughtful suggestion. We understand your concerns regarding the baseline assumption of biomaterials’ carbon storage and the implications of such materials’ end-of-life. However, we have decided to keep MNETon as our default scenario based on our careful analysis and understanding of how biogenic carbon is stored. As a complement to Comment 13, our perspective is that bioplastics, unless incinerated or recycled, can indeed serve as a carbon sink. This is largely due to drop-in plastics’ properties such as durability and resistance to corrosion; while contributing to their notoriety as pollutants, these properties also suggests that plastics – whether fossil- or bio-based – do not release their stored carbon. While we agree that a very significant share of plastics “have very short lifetime”, their carbon will not be released unless it is recycled or incinerated, which is not the case for most regions so far. The carbon removal results in bioplastics from scenarios with MNETon should not be interpreted as carbon sink in landfills but rather the potential of carbon sink in non-recycled and non-incinerated plastics.
Notes/actions	As a complement to Comment 13 actions, we also added the following sentence in the Discussion: Although decarbonisation technologies are available and could reduce emissions to as low as -1 GtCO₂yr⁻¹ by 2050, these reductions heavily rely on the large-scale availability of alternative feedstocks and on accounting for bio-based carbon storage in bioplastics that are neither incinerated nor recycled. In scenarios where alternative feedstocks is limited and carbon storage in biomaterials is not assumed, achieving net-zero emissions in the chemicals sector by 2070 remains elusive.

Comment 19	19. L435: What is the minimum deployment of CCS and how does it compare with your assumption of 10 Gt/year? I think 10 Gt/year CCS is still quite high, especially compared with the 45 Mt/year level we are at today.
Response	We thank the reviewer for the question about the deployment of CCS. We plotted below the total CO₂ captured globally for each scenario. It includes DAC, Fossil CCS, BECCS, and CCS of process emissions across all sectors, including the chemical sector. We list a few points about these trajectories:  • Scenario NP_i represents the minimum deployment of CCS, which is zero across all years. • Scenarios 1p5C_gCCS and 1p5C_all follow strictly the constraint implemented in CCS, i.e., CCS is used to the limit. • When not restricted (scenarios 1p5C, 1p5C_PBIO, and 1p5C_MNEToff), CCS scales up faster in the 2025-2050 period. It

further increases to up to 15000 MtCO₂.yr⁻¹ in scenarios 1p5C and 1p5C_MNEToff. When we restricted biomass in scenario 1p5C_PBIO, however, we did not see this increase after mid-century. In this case, CO₂ captured converged to the 10000 MtCO₂.yr⁻¹ levels. This indicates that biomass use is highly associated with CCS after mid-century in mitigation scenarios.

We use in our study the same constraints limiting biomass and upscaling CCS defined in recent model intercomparison protocols. In these exercises, CCS upscaling was based on recent estimates of CCS projects (Consoli C, 2018) and then assuming 15% yearly growth rate. Similar approaches and calculations in the literature were also included to justify the levels suggested as constraints in scenario development (Haszeldine et al., 2018).

In the SI, we also refer to the CCS levels in C1,C2, C3 scenarios of AR6 database.

References:

Consoli C (2018) CCS Storage Indicator (CCS-SI). Global CCS Institute: Melbourne, Australia
 Haszeldine RS, Flude S, Johnson G, Scott V (2018) Negative emissions technologies and carbon capture and storage to achieve the Paris Agreement commitments. Philosophical Transactions of the Royal Society A: Mathematical, Physical and Engineering Sciences 376:20160447. <https://doi.org/10.1098/rsta.2016.0447>

Notes/actions

N/A

Comment 20

20. L442: Does this 100EJ/year limit set for all years? I think this is also too high for the sensitivity scenario. Is it possible to set the limit at the lower bound of IAM scenarios? Also, what are the considered biomass sources? What are the other biomass consumers besides chemical sector?

Response	We thank the reviewer for the question about the 100EJ.yr⁻¹ limit for scenario 1.5C_PBIO as well as the biomass sources and sectors that use biomass. Similarly to the limits implemented to CCS in scenario 1.5C_gCCS, this value was taken from recent model intercomparison protocols, as mentioned in the scenario development subsection in the online methods. While there is a wide literature covering large ranges of the technical potential of primary biomass use, Creutzig et al. (2015) find a high agreement on levels up to 100EJ/yr globally for sustainable production. The biomass sources included in COFFEE primarily include planted forests and energy crops. Maize, sugarcane, beet, soybean, eucalyptus, animal fat, wheat, and grassy biomass are raw materials converted into solid and liquid biofuel (e.g., wood chips/pellets, bioethanol, HVO, HEFA, biodiesel, green diesel, and co-products) via thermochemical and biochemical routes. Some of these processes can be equipped with CCS. Sectors that use commercial biomass include not only the chemical sector but also: 1) the transportation sector: for passenger vehicles (bioethanol, biodiesel, green diesel/HVO) and aviation (HVO/HEFA) and freight trucks as well as shipping (biodiesel, HVO, HEFA, bioethanol, biomethanol); 2) cement: charcoal and bio-waste can be used to supply high-temperature heat 3) blast furnaces: charcoal can be used as a reductant; 4) other industries – which does not explicitly but implicitly represents the paper and pulp sector and its biomass demand. 5) electricity – biomass power plants. While biomass can be used in all those sectors, it will not necessarily be used in these sectors. However, bio-based pathways are included as options to fulfil those distinct demands in mitigation pathways, which, therefore, might compete for its use. References: Creutzig F, Ravindranath NH, Berndes G, et al (2015) Bioenergy and climate change mitigation: an assessment. GCB Bioenergy 7:916–944. https://doi.org/10.1111/gcbb.12205
Notes/actions	We expanded the section about scenario development to add the abovementioned references justifying the use of 100EJ/yr as a limit to sustainable primary biomass use. Scenario with Restricted Use of Biomass (PBIO) Based on the AR6 scenario database, biomass use as primary energy ranges from 28.7 to 228.4 EJ.yr⁻¹ in 2030, 33.5 to 310.1 EJ.yr⁻¹ in 2050, and 41.1 to 530.4EJ.yr⁻¹ in 2100 (Figure S5 in the SI). The Restricted Use of Biomass scenario (PBIO) considers a world where the use of biomass is restricted to below 100 EJ.yr⁻¹, based on the high-confidence agreement in the literature found in ref.45 for the sustainable technical potential of primary biomass production. We also included a section in the SI where we detail the sources of carbon feedstock, including bio-based feedstocks, as follows: S2.1. Sources of carbon feedstock COFFEE represents fossil-, bio- and CCU-based feedstocks as options for producing primary chemicals.

Fossil-based feedstock: Primarily includes natural gas (i.e., methane and ethane), oil (i.e., LPG, naphtha and heavy oil), and coal. Natural gas can be produced in fields associated with oil or in dedicated fields. Ethane and methane can be recovered from natural gas and used for steam cracking and steam methane reforming units, respectively. LPG, naphtha and heavy oil are outputs of the petroleum refining sector, which is described in ref.¹². LPG is an input to propane dehydrogenation process, naphtha can be used in steam cracking and catalytic reforming units whereas heavy oil is used for hydrogen to ammonia production via partial oil oxidation. Coal is produced in different qualities in the resource module (bituminous, subbituminous, and lignite) but only bituminous coal is allowed in coal gasification processes.

Bio-based feedstock: Primarily includes solid and liquid biofuel produced from planted forests and energy crops. Solid biofuel is obtained from woody (eucalyptus) and grassy sources as well as agriculture residues, including sugarcane bagasse. Eucalyptus can be used for biomass gasification processes to produce hydrogen (for energy or ammonia production), methanol, or can be used in Fischer-Tropsch Biomass-to-Liquids (FT-BtL) to produce advanced liquid biofuels. Sugarcane bagasse can be used for 2G bioethanol production.

Liquid biofuels included as feedstocks for the chemical sector are bioethanol (via fermentation/distillation from sugarcane, maize, wheat, sugar beet, and agricultural residues), and bio-naphtha, and bio-LPG as co-products of hydrotreated vegetable oils (HVO) or hydroprocessed esters and fatty acids (HEFA) production from soybean oil, maize oil, or animal fat. HVO and HEFA are advanced biofuels alternatives to be used in diesel engines as green diesel or jet fuel (or sustainable aviation fuel – SAF), and might co-produce bio-naphtha and bio-LPG to be used in other sectors, including as feedstocks.

Biomethane could be a promising alternative feedstock. However, its supply chain is not yet developed in COFFEE and it remains as a topic to be further explored in future work.

CCU-based feedstock: CO₂ as a feedstock is sourced from a “CO₂ pool”, which is composed by CO₂ captured from: 1) fossil- or bio-based concentrated sources such as power plants, cement kilns, blast furnaces, and methanol/ammonia plants (via gasification/partial oxidation, but not electrolysis). Both energy and process emissions can be captured from concentrated sources, and costs for CO₂ transportation are considered; 2) diluted CO₂ in the atmosphere via Direct Air Capture (DAC) based on absorption in sodium hydroxide solution, based on refs,^{13,14}. Captured CO₂ feeds into a regional “CO₂ pool” in the model, where the model decides whether it is geologically stored or used for CCU.

Reviewer #2:

This paper assessed the refinery sector with an explicit representation of basic chemicals under climate change mitigation scenarios by using an Integrated assessment model named COFFEE. I think this paper is premature, and its quality is insufficient to be published as a paper in Nature Communications. While I believe the attempt would be worthwhile, the research objective is unclear; easy mistakes can be seen, and the paper just shows primary model outputs rather than elaborated analysis.

The major comments are as follows:

We thank Reviewer #2 for the careful review as well as for encouraging us to provide additional explanations and analyses. Your critical insights have helped refine our paper significantly to ensure a good reading flow and to clearly state a research question in the introduction. We believe that the revisions made in response to these comments, which are addressed one by one below, have greatly improved our manuscript.

Comment 1	The research objective needs to be clarified. This is really critical because the reviewer cannot assess whether the conclusion is appropriate or not. This would lead to further following essential problems
Response	We appreciate the critical feedback, which prompted us to reconsider our approach and led to significant improvements in the formulation of our research question. While all the elements we touch upon were mentioned and considered in the introduction, we agree that it was not fully clear and that the main messages needed to be highlighted. Our main contribution with this work is to emphasize that fossil fuels phase-out/phase-down policies must target both energy and non-energy-uses given its intertwined nature; otherwise, chemical demand will drive refineries' utilisation factor up naturally driving fuel costs down and leading to refineries being shut down later than planned. Therefore, the issue of the chemical sector as a major consumer of FF for non-energy purposes is not only to be 'hard-to-abate' as it is commonly mentioned in the scientific literature but also 'hard-to-defossilize'. We believe this is the first time a global IAM addresses this issue.
Notes/actions	We reframed the introduction to clearly state the hard-to-abate and hard-to-defossilize problem and to pose the following research question: Despite the decreasing demand for fossil fuels in energy systems due to climate policies^{3,20,21}, the co-production of energy, feedstocks, and chemicals in petroleum refineries and implications to energy use and emissions pathways across sectors remains largely unexplored. Aiming to fill this gap, this article investigates the role of the chemical sector in a global net-zero strategy. Our hypothesis is that the chemical sector, while being both hard-to-abate and hard-to-defossilize, can offer strategic contributions to deep decarbonisation, both sectoral and systemically. We argue that the interdependent relationship between fuels and chemicals means that efforts in the chemicals sector are relevant not only to achieving sectoral net-zero but also to reducing the utilisation factor of petroleum refineries. This leads to indirect lower fossil fuels use and mitigation across all sectors. Moreover, as alternative routes for chemicals rely on the supply of biomass, H₂ and CO₂, results from an integrated assessment perspective should reveal potential competition for these resources between energy and non-energy uses and between carbon capture and utilisation (CCU) and CCS. We also reformulated the discussion to directly relate to the introduction as follows: This work contributes to the research field of carbon emissions mitigation in the primary chemicals sector in four ways. Firstly, we observe that primary chemicals are hard-to-abate but can have a role in promoting systemic decarbonisation. Although decarbonisation technologies are available and could reduce emissions to as low as -1 GtCO₂yr⁻¹ by 2050, these reductions heavily rely on the large-scale availability of alternative feedstocks and on accounting for bio-based

carbon storage in bioplastics that are neither incinerated nor recycled. In scenarios where alternative feedstocks are limited and carbon storage in biomaterials is not assumed, achieving net-zero emissions in the chemicals sector by 2070 remains elusive.

Secondly, primary chemicals are hard-to-defossilize, leading to critical implications for decarbonisation of the chemicals sector and beyond. Our findings suggest that ambitious feedstock substitution could reduce fossil fuel dependence by approximately 60% by 2050 globally. However, if carbon capture and biomass use are constrained, the feedstock use profile remains essentially unchanged. This reflects both an opportunity and a challenge. On the one hand, substituting feedstock can significantly reduce refinery utilisation bringing indirect decarbonisation across sectors. On the other hand, failing to transition away from fossil fuels use as feedstock could inadvertently sustain petroleum refining activities, prolonging fossil fuel reliance and delaying the broader transition away from fossil fuels. This scenario could lead to continued fossil fuel extraction and refining focused on feedstock production, potentially resulting in lower-cost fuel co-production, and assets becoming stranded before investments are amortised. These findings highlight critical implications for investments in fossil fuel assets and underscore the need for a more integrated approach to energy, climate, and resource regulation.

Thirdly, transitioning away from fossil fuels will require restructuring within the primary chemicals sector. Steam cracking remained the leading technology for HVC production over the century, largely influenced by the electrification of the transport sector and the resulting diversion of gasoline to the naphtha pool. However, refinery-sourced HVCs will decline due to the shrinking of the refining sector as a result of fossil fuels phase-out policies. On-purpose routes, particularly the ones based on methanol (via MTO, MTA) and ethanol (via BDH, and also influenced by oversupply due to transport electrification) as intermediates, will increase to accommodate feedstock substitutions. The restructuring will also affect methanol and ammonia production; biomass gasification with CCS becomes the preferred route for ammonia, while CCU plays a key role in methanol production. Demand for non-energy methanol in mitigation scenarios is expected to increase two to five times by 2050 compared to NPi. This is primarily due to its importance as an intermediate for HVC production. This shift indicates that solutions are product-specific and every carbon feedstock source will be relevant for achieving climate targets.

Importantly, our scenarios explore a limited set of technologies to understand the role of feedstock supply in achieving net-zero scenarios, assuming unchanged demand patterns for primary chemicals. Given the diverse products and services provided by primary chemicals, detailing and addressing their complexities is challenging. This work aimed to unravel the dynamics of petroleum production and refining within the emissions pathway of primary chemicals by developing drop-in substitution alternatives. Nevertheless, we acknowledge that more is needed to provide a comprehensive view of the potential futures lying ahead for the chemical sector. Historically, energy transitions have redefined systems and triggered material transitions and vice-versa³⁶. Hence, transforming the complex network of feedstocks, products, and services developed under the petroleum-based economy and thermochemical foundations since the 1950s will require innovative solutions. Beyond phasing-out fossil fuels in energy systems, these include the development of biodegradable chemicals, material substitutions, and demand-side measures. While these aspects were not

	the focus of our assessment, they are crucial considerations for future policymaking. Fourth and lastly, improving the representation of fossil fuels in IAMs to capture relevant material-energy links allows modellers to reach a higher degree of accuracy in depicting systems integration. Representing incumbent energy carriers set to be phased-out in detail enables modellers to provide realistic but ambitious recommendations to policymakers considering their pervasive use across sectors. In this sense, the scope and scale of the required transition in the chemical sector to respond to the climate challenges of the 21st century are diverse and in different stages of the supply chain.
--	---

Comment 2	The abstract and conclusion can mainly be derived without having a model simulation. For example, the following statement taken from the abstract is obvious. What are the other alternatives? “achieving more than 50% of oil use reduction in 2050 requires a comprehensive feedstocks and material substitution strategy on top of electrification and fuel switch to offset ceased output from global oil refining processes.” Similarly, in the other sentence of the abstract, I have no alternative idea of how to phase out oil, which is obvious. “we estimate that without materials efficiency measures and timely scale-up of alternative feedstocks, oil phase-out is unfeasible”
Response	We thank the reviewer for this comment. With the sentences quoted directly from the previous abstract, we aimed at emphasizing that without fossil fuels phase-out as feedstock, the complete fossil fuels phase-out – both in energy and non-energy systems – is unfeasible. We agree that this was not very clear and we improved the abstract to reflect the value of this study.
Notes/actions	We re-wrote the abstract to highlight more specific findings of our study, as follows. The discussion/conclusion was also improved (see Comment 1 above). FROM: Around 13% of fossil fuels are used for non-combustion purposes globally. Fossil fuel processing plants, such as petroleum refineries, present an interdependent material and energy systems dynamics. Hence, without addressing the non-energy output in fossil fuel plants, decarbonisation of the energy system becomes even more challenging. This study explores the future role of fossil fuels use for non-energy purposes in climate and alternative feedstocks stringent scenarios, with a focus on the basic chemicals sector. By using a global integrated assessment model (IAM) that features a detailed refining and basic chemicals sectors, we estimate that without materials efficiency measures and timely scale-up of alternative feedstocks, oil phase-out is unfeasible. Our findings indicate that achieving more than 50% of oil use reduction in 2050 requires a comprehensive feedstocks and materials substitution strategy on top of electrification and fuel switch to offset ceased output from global oil refining processes. TO: Around 13% of fossil fuels globally are used for non-combustion purposes. Fossil fuel processing plants, such as petroleum refineries, exhibit interdependent material and energy system dynamics, making the transition away from fossil fuels energy systems becomes more

challenging without addressing the non-energy outputs. This study explores the future role of fossil fuels for non-energy purposes in climate-stringent scenarios with restrictions on alternative feedstock availability, focusing on the primary chemicals sector. Using a global integrated assessment model with detailed refining and primary chemicals sectors, findings reveal that up to 62% of fossil fuels used as feedstock in the chemical sector could be substituted by alternatives by 2050 with significant scale-up of biomass utilisation and carbon capture technologies. Annual CO₂ emissions from the chemical sector could be reduced to as low as -1Gt CO₂ by the same year if carbon storage in non-recycled and non-incinerated bioplastics is accounted for.

Comment 3 There is no description of DAC and hydrogen-related things, which look like critical technology. It is because alternative hydro-carbon (biofuel or synthetic fuel) is the only way to phase out fossil oil, and that means carbon source technology and hydrogen are key. Moreover, hydrogen potential and cost are relevant to the variability of renewable energy power generation and its adjustment role. The hydrogen cost would directly change the condition of synthetic fuel cost, which is highly relevant to the electricity dispatch. However, the current model seems to be unaddressed that part.

Response We thank the reviewer for the questions raised about DAC and hydrogen.

We agree with the reviewer on the relevance of DAC and hydrogen technologies as well as their costs given how the renewable power is modelled. Although hydrogen techno economic assumptions are described in the SI, DAC was not fully described in the SI and Online methods because it is a carbon removal technology that is not represented within the chemical sector. DAC has a link with the chemical sector via a “CO₂ pool”. We clarify this in the bullet points below:

- 1. Hydrogen can be produced via various routes.** Water electrolysis (using both renewable and non-renewable electricity), biomass gasification (w/ and w/o CCS), coal gasification (w/ and w/o CCS), partial oxidation of oil and steam methane reforming (w/ and w/o CCS). It is also co-produced in steam cracking processes and catalytic reform of naphtha. COFFEE does not apply constraints in electricity generation. Thus, the model can deploy conventional or alternative electricity generation technologies to 1) Fulfil electricity demand in the current level of participation, 2) Electrify further energy services and 3) Produce hydrogen or store electricity in batteries.
- 2. CCU and CCS compete for the same carbon dioxide sources in COFFEE:** There are two pathways through which CO₂ as a feedstock is sourced to the chemical sector in COFFEE. 1) From fossil- or bio-based concentrated sources such as power plants, cement kilns, blast furnaces, and methanol/ammonia plants (via gasification/partial oxidation, but not electrolysis). Both energy and process emissions can be captured from concentrated sources, and costs for CO₂ transportation are accounted for, although in a stylized and simplified manner. 2) From diluted CO₂ in the atmosphere via Direct Air Capture (DAC). Captured CO₂ thus feeds into a “regional pool” in the model, where the model decides whether it is geologically stored or used for CCU.

	3. COFFEE does not apply constraints in electricity generation. Thus, the model can deploy conventional or alternative electricity generation technologies to 1) Fulfil electricity demand in the current level of participation, 2) Electrify further energy services and 3) Produce hydrogen or use batteries to store electricity. Therefore, the cost of green hydrogen is highly dependent of the electricity cost, which is endogenously calculated. 4. In very ambitious carbon stringent scenarios, the model decision weights towards negative emissions: not only CCU via DAC or other carbon sources are more expensive than other carbon sources, but they also reduce their participation in highly ambitious scenarios because BECCS or other CDR measures become critical to stay within the carbon budget. Even so, in stringent scenarios, DAC deployment reaches levels of ~700MtCO₂ by 2100 globally (or 16% of all carbon captured), indicating it can be a relevant carbon source where biomass is not available. To sum up, the model is not trying to find the optimal solution for the chemical sector or to provide the pathway that leads to a chemical industry net-zero by 2050. In our case, from a linear-optimization integrated assessment perspective, the model aims for a cost-optimal solution of achieving global temperature goals in the whole system, including the transport, electricity, buildings, industry, and land-use sectors. This leads to competition for resources in those sectors. Therefore, resources are best allocated when they lead to overall emissions reduction. Future studies focusing on how policy incentives could lower the costs of CCU would be valuable. For instance, integrating electrolysis-based hydrogen production with the use of oxygen produced as a by-product in oxy-fuel processes could result in a more concentrated CO₂ stream from combustion processes. This concentrated CO₂ is easier to capture and purify for further utilisation in the chemical sector.
Notes/actions	We expanded the methods section to include a description of carbon sources in COFFEE, which includes DAC. Techno economic data for hydrogen production is also described in the SI in the ammonia section, and we added a paragraph to indicate that those parameters were also used to represent hydrogen for energy purposes. Table S3: Techno-economic assumptions for hydrogen and ammonia production technologies. SMR, SMR+CCS, CGS, CGS+CCS, POX, BGS, BGS+CCS, and Electrolysis are hydrogen production technologies, which can be used for energy and non-energy purposes (i.e., in ammonia production via Haber-Bosch reaction).

Comment 4	The overall scenario experimental design is not thoroughly considered. As it has already been pointed out above, the research objective is unclear, and I am not sure how to judge the appropriateness of the scenarios, but let me suppose that the authors wanted to question how much alternative material is needed to achieve a 1.5-degree target for the chemical sector. Then, obviously, the current scenario cannot be answered because CDRs can offset it anyway without changing oil-based petrochemistry under some conditions but such kind of things are not addressed. This is just an example, but either way, without an essential research question, there is no way to judge.
------------------	---

Response	We thank the reviewer for the comment. As mentioned above, we included a clearer RQ: “this article investigates the role of the chemical sector in a global net-zero strategy. Our hypothesis is that the chemical sector, while being both hard-to-abate and hard-to-defossilize, can offer strategic contributions to deep decarbonisation, both sectoral and systemically. “ Therefore, we designed scenarios compliant with 1.5-degrees increase to understand the role of the chemical sector as hard-to-abate; the sensitivity MNEToff adds in this discussion so we can identify the role of carbon removal in biomaterials in net-zero goals for the chemical sector. We designed other sensitivities to limit feedstock availability (gCCS and PBIO) and test to what extent the chemical sector is hard-to-defossilize and draw insights to how this affects emissions pathways in the chemical sector and in the whole system. Important to highlight that these feedstock sources are also mitigation measures in other sectors, so we are also testing the role of alternative feedstock availability in competition with other mitigation measures (i.e., store vs use CO₂; biomass use vs BECCS). The scenario all combines restrictions on emissions and feedstocks so we can analyse the role of FF in fulfilling chemicals demand. It is also important to stress that we do not analyse the chemical sector in isolation, but integrated to other systems and subject to global/regional constraints. Therefore, the research question is more related to “what is the role of chemicals in emissions reduction and fossil fuels use” rather than “how can we drive down emissions and fossil fuel use in the chemicals sector”.
Notes/actions	The introduction and scenario design sections were rewritten to enhance clarity.

Comment 5	The techno-economic assumptions in the supplementary information are impossible to validate. I have gone through the supplementary tables, which show key parameters and source references. I also went through some of the references, but I could not easily figure out how and from what information they are assumed (e.g. technology CR Bionaphtha)
Response	We thank the reviewer for this comment and concern about transparency of assumptions about techno economic data. Several steps were taken to enhance the clarity and verifiability of our methodologies, which are explained below.
Notes/actions	We added a new subsection in the Extended Methods to provide details about the parametrization process: S2.2.4. Technoeconomic parametrization of chemical processes Table S1, Table S2, and Table S3 detail the technoeconomic parameters for the production of HVCs, methanol, and ammonia, respectively. Primary references were used for refinery^{41,63} and chemical processing^{3,64} units data, which were validated and complemented drawing on multiple sources. To ensure consistency and comparability across processes, we standardized all values to the same units and currency. This process included the conversion of economic and physical units. For economic conversions, the Chemical Engineering Plant Cost Index (CEPCI)⁶⁵ was used to adjust the investment costs of chemical plants to 2010 US dollars. This accounts for inflation and variations in the costs of equipment, materials, and labour over time. Moreover, due to the diversity in original

	data – often presented in various currencies and/or from different years – we performed extensive conversions and validations. For physical conversions, we used densities, heat of combustion, and basic unit conversion for handling material and energy efficiencies. When using heat of combustion, we applied the Higher Heating Value (HHV) when steam was explicitly included in the exhaust gases, capturing the total energy released, including the latent heat of vaporization of water. Conversely, we used the Lower Heating Value (LHV) when steam was not explicitly represented, which omits the energy associated with water vapor condensation. This differentiation ensures accuracy in energy conversion calculations depending on the presence of steam in the combustion exhaust. In our analysis, we treated units that process bio-based, CCU-based, and fossil-based feedstocks as equivalent. For example, we assumed that catalytic reforming units convert both naphtha and bio-naphtha with identical efficiency. Similarly, whether derived from fossil, CCU or bio-based processes, methanol is chemically identical. Therefore, we maintained consistent costs and yields across all these units regardless of the feedstock origin.
--	--

Other minor:

Comment 1	Why does ammonia need BECCS? Presumably, ammonia does not require carbon and does not need to be BECCS. As described, hydrogen is the essential source, which I intuitively think is available from electrolysis.
Response	We thank the reviewer’s insightful questions. Although ammonia production itself does not require carbon, the results of our study highlights an advantage in using biomass gasification w/ CCS (BGS+CCS) for two primary reasons:  1) BGS+CCS enables easier-to-capture bio-based CO₂. The process produces concentrated CO₂ and hydrogen; while hydrogen is extracted to react with N₂, CO₂ is fully captured. In comparison, CO₂ captured in methanol production is less pure because methanol inherently requires carbon. Because 1.5 scenarios are becoming extremely difficult to achieve, our analysis suggests that model decisions between ammonia and methanol production are influenced by the costs associated with carbon storage rather than carbon utilisation. 2) Direct utilisation in Urea production. There is a specific demand for pure CO₂ in urea production derived from ammonia processes. Even today, and since ammonia production scaled up, coal and gas are the main feedstocks in the hydrogen production steps. While not necessarily the most efficient approach for the ammonia sector alone, BGS+CCS presents a strategic advantage within an integrated assessment of the entire system for very stringent decarbonisation scenarios. While water electrolysis is carbon-neutral and green hydrogen fit for some applications, biomass gasification with CCS is identified as a cost-effective alternative to reduce not only sector but global emissions. Further studies should be performed to see if this findings change when we integrate electrolysis-based hydrogen production with the use of oxygen produced

	as a by-product in oxy-fuel processes. This could result in a more concentrated CO ₂ stream from combustion processes. This concentrated CO ₂ is easier to capture and purify for further utilisation in the chemical sector.
Notes/actions	N/A

Comment 2	There is no distinction in the CO ₂ sources between fossil combustion, biomass combustion or DAC. How is it modeled? CO ₂ can be taken from fossil capture, biomass capture, or atmosphere.
Response	We thank the reviewer for this comment. As mentioned above, there are two pathways through which CO ₂ is sourced as a feedstock to the chemical sector in COFFEE. 1) From fossil- or bio-based concentrated sources such as power plants, cement kilns, blast furnaces, and methanol/ammonia plants (via gasification/partial oxidation, but not electrolysis). Both energy and process emissions can be captured from concentrated sources, and costs for CO ₂ transportation are accounted for, although in a stylized and simplified manner. 2) From diluted CO ₂ in the atmosphere via Direct Air Capture (DAC). Captured CO ₂ thus feeds into a “regional pool” in the model, where the model decides whether it is geologically stored or used for CCU in chemical production.
Notes/actions	Based on this and other reviewers’ concern, we added a section in the SI file to describe how the carbon feedstock sources are modelled in COFFEE. A shorter section was added to the Methods section in the Manuscript.

Comment 3	The hyperlinks to figures are not working.
Response	We apologize for the formatting issues. These errors likely occurred during the automatic conversion to PDF format within the online submission system.
Notes/actions	To avoid these issues, we now pre-generate the pdf prior to submission.

Reviewer #3:

Summary

- This is an interesting paper, probably the first I have seen using an IAM, that considers chemical products in a netzero context in detail.
- There are numerous English grammatic errors.
- The referencing system is broken, and I was not able to check claims against the references.
- The paper is interesting, but feels underdeveloped.
- Summary: Major revision.

Editorial

- Minor English issues, e.g., “present an interdependent material and energy systems dynamics” in the abstract should be “present an interdependent material and energy systems dynamic” or “present interdependent material and energy systems dynamics”. I won’t read for English or grammar issues further, but the authors should do a thorough review.

We thank reviewer #3 for the balanced and careful review. We are thankful for the time taken to review our paper and we apologize for our English grammar errors. We have carefully

revised the entire paper and we respond to specific comments one-by-one, detailing the actions we have taken for each. This feedback has been invaluable in guiding these revisions.

Substantive:

Comment 1	Page 3 lines 61-70. Careful about being overdescriptive of the feedstocks for chemicals. In the main China uses coal and some naphtha for most feedstocks, Europe uses mainly crude oil derived naphtha, and North America C5+ liquids from natural gas, and gas directly. The relative feedstock costs vary.
Response	We thank the reviewer for pointing that out. We agree that key regional differences might have gotten lost in our general overview of feedstocks, although we carefully considered it in the modelling and data collection phases (actually, this was a major challenge for the global IAM improvement as well). It was a challenge to write the paper's introduction to summarize the problem for climate change mitigation experts but non-experts in the chemical sector (and vice versa) while dealing with the complexity of the sector, e.g., integration with the refining sector, multi-product processes, and particularities of methanol, ammonia, and HVCs. We cannot do this topic enough justice in the introduction, given the limited space, but when describing the technologies considered in the SI, we mention a few of these regional specificities regarding chemical feedstock. For example: "...coal gasification (CGS) has been prioritized in countries with vast coal reserves and where the transportation of natural gas is not economical, such as China and South Africa " "Propane dehydrogenation (PDH) is one of the main technologies used so far to bridge the propylene supply gap, mainly in the United States, Middle East, and China."
Notes/actions	We changed that paragraph to include some regional specificities: With the demand for fossil fuels in energy systems projected to decrease³, the materials systems will also be affected primarily because fossil fuel plants simultaneously co-produce fuels and feedstock for materials production. Naphtha, the main feedstock used to produce high-value chemicals (HVCs) (i.e., ethylene, propylene, butadiene, and aromatics), is inexpensively co-produced with diesel, gasoline, and aviation fuels. Refinery units directly co-produce propylene⁴ and aromatics⁵ while producing fuels. Natural gas (for example in North America and the Middle East) and coal (mainly China and South Africa) are also key feedstocks for producing not only HVCs but also methanol⁶ and ammonia⁷.

Comment 2	Page 5-65, line 16-149 This sentence is confusing "gasoline and naphtha are hydrocarbons that share a similar carbon range, passenger vehicle electrification leads to the use of gasoline streams to meet feedstocks increasing demands in the basic chemicals sector". Do you mean "increasing feedstock demands in the basic chemicals sector", ie gasoline replaces naphtha?
Response	We thank the reviewer for pointing out the confusion in this sentence.
Notes/actions	We rephrased it to make it clear.

	FROM: “gasoline and naphtha are hydrocarbons that share a similar carbon range, passenger vehicle electrification leads to the use of gasoline streams to meet feedstocks increasing demands in the basic chemicals sector”. TO: “This happens simultaneously with the electrification of passenger vehicles, leading to gasoline oversupply. As gasoline and naphtha share a similar carbon range, gasoline replaces naphtha as feedstock for HVCs.”
--	---

Comment 3	Line 207 “Error reference not found” other after it
Response	We apologize for the formatting issues. These errors likely occurred during the automatic conversion to PDF format within the online submission system.
Notes/actions	To avoid these issues, we now pre-generate the pdf prior to submission.

Comment 4	Line 222 Explain more carefully “This results in a higher share of on-purpose routes deployment (Error! Reference source not found.b) to bridge the propylene gap caused by a product split more selective to ethylene than other HVCs in ethane compared to Naphtha S”
Response	We thank the reviewer for this suggestion, we agree that this sentence is confusing and requires clarification.
Notes/actions	We changed that paragraph as described below: FROM: Our results show that steam cracking (SC) remains the main technology to produce HVCs. Ethane SC increases from 9% in 2010 to 25% of HVC production in all mitigation scenarios in 2030. This results in a higher share of on-purpose routes deployment (Error! Reference source not found.b) to bridge the propylene gap caused by a product split more selective to ethylene than other HVCs in ethane compared to Naphtha SC (Figure S3 in the SI). TO: Our results show that steam cracking (SC) remains the leading technology for producing HVCs. Ethane SC increases from 9% to 25% of HVC production in all mitigation scenarios between 2010 and 2030, particularly in gas-rich regions such as the Middle East and the US. Ethane SC has a higher selectivity towards ethylene over other HVCs, compared to Naphtha SC (Figure S2 in the SI). As a result, switching from naphtha to ethane as a feedstock leads to propylene and aromatics gaps in demand/supply ratios. Furthermore, as the passenger transport sector gradually electrifies, refineries reduce their utilisation factors and their HVC output (Figure 3-b). This results in a gasoline surplus, which becomes cheaper and finds market in the primary chemical sector through Naphtha Catalytic Cracking (NCC) (32-52% of HVC market share by 2070). NCC has a more balanced ethylene/propylene ratio than conventional Naphtha SC³⁴ (see SI) – and catalytic reforming units. In scenario all, the share of electricity use in the transport sector reaches its highest level (37% on final energy use basis considering passenger and freight transport in 2070) as biofuels reduce their importance for BECCS given CCS and biomass constraints. Moreover, reduced HVC output from refineries makes the propylene and aromatics gap more pronounced. NCC deployment is not sufficient to bridge these gaps alone and on-purpose routes such as Propane

	Dehydrogenation (PDH), Metathesis (MTT), and Methanol-to-Olefins (MTO) see substantial growth (Figure 3-b).
--	--

Comment 5	Line 242 Transport electrification maxes out at 37%? This seems on the face of it very low, please explain. Total transport demand for fossil then liquid fuels is critical to your results.
Response	We thank the reviewer for the insightful question. The 37% electrification rate refers to both freight and passenger transport. Thus, we are also including maritime and aviation transport here, as well as cargo trains and trucks, which are somewhat harder to electrify. Furthermore, this percentage is based on final energy use. While this may appear modest from an energy perspective, it is significant given that electric vehicles are highly efficient. Consequently, the impact in terms of energy service provision (pkm/tkm) is substantially greater.
Notes/actions	We clarified in the manuscript that 37% refers to final energy use: “(37% on final energy use basis considering passenger and freight transport in 2070)”

Comment 6	Line 263 You only get negative emissions with biomass sources for hydrogen if permanent CCS is applied to the carbon, or the carbon is locked permanently in a product (Tanzer & Ramírez, 2019)
Response	We agree with the reviewer's statement, and we clarified the referred sentence below.
Notes/actions	FROM: Results on syngas products show that biomass gasification is prioritized for ammonia production due to the potential for BECCS whereas CCU is prioritized for methanol production (Error! Reference source not found. d, e) This happens because, in ammonia production, hydrogen is the primary product, leading to either negative emissions with biogenic sources or emission reductions with fossil sources, depending on the extent of carbon capture. However, water electrolysis coupled with O₂ use to capture CO₂ via oxyfuel routes might also be promising. For methanol production, part of the carbon content remains in the material. Overall, the use of fossil fuels for ammonia reduces over time but remains relevant, especially in more restrictive scenarios. TO: Results on syngas products show that biomass gasification with CCS is favoured for ammonia production due to the potential for BECCS, whereas CCU is preferred for methanol production (Figure 3 d, e). While in ammonia production carbon is largely converted into captured CO₂ with high purity, contingent on the efficiency of the capture process, in methanol production, some of the carbon remains in the final product. Therefore, scaling-up biomass use with carbon capture in ammonia production is identified by the model as a source of negative emissions. This can significantly reduce emissions across the overall system both in scale and pace required for reaching 1.5-degrees goals. Nevertheless, water electrolysis coupled with O₂ use in oxyfuel routes for capturing CO₂ could also be effective, although not assessed in this work. Overall, the use of fossil fuels for ammonia production reduces over time but remains relevant, especially in more stringent scenarios.

Comment 7	Line 272 The degree of net neutrality of biomaterial storage of carbon is conditional on its longevity, e.g., most plastics are burnt at end of life, which would make them atmospheric neutral depending on the biomass species, how it was harvested, and what happened with the soil carbon as result of harvesting. My understanding is this is not as much of an issue in Brazil, where there is a lots of agricultural waste, but it can be big problem in temperate climates where there are with forest tradeoffs and soil carbon loss. (Hepburn et al., 2019)
Response	We thank the reviewer for the comment concerning the end-of-life emissions of biomaterials as well as the system definition. We elaborate on those in the bullet points below:  • COFFEE, as an integrated assessment model, primarily accounts for direct emissions within each sector. It does not explicitly calculate life-cycle emissions for individual products. Instead, it comprehensively represents all sectors and their interconnected supply chains, explicitly allocating direct emissions across each sector. Therefore, the land-use sector is represented and linked to the chemical sector, hence indirectly considering the impacts of land-use changes. Emissions resulting from energy use and land-use carbon balance taking place during the cultivation and harvesting of biomass used for food, energy, or feedstock purposes are allocated into the land-use module. • COFFEE accounts for emissions according to the End-of-Life of HVCs. Current statistics show that plastic landfilling and mismanagement sum up to 71% worldwide. We used country-level data aggregated to the regional level based on current levels of incineration, landfilling/mismanagement, and recycling using the World Bank’s ‘What a waste’ database. We assumed that these levels remained constant over the century, which we acknowledge as a limitation of the study to be addressed in future studies. Emissions for incineration and recycling are taken into account. Therefore, while Europe, Korea, and Japan have a high incineration rate (and those emissions are accounted for), that is rarely true for other regions. As landfilling is still relevant, all the carbon that has this destination by the end of its lifetime is considered to be stored. This is done for carbon accounting reasons, not to support landfilling. The reason why we have a plastic accumulation and pollution problem is precisely because of its stability and non-degradability. Hence our modelling approach aims to capture that effect as well. This approach was similarly adopted by Stegmann et al. (2022).
Notes/actions	We included two sections as a response to this and other reviewers’ comments on the sources of carbon feedstock and final disposal assumptions in the SI: S2.1. Sources of carbon feedstock COFFEE represents fossil-, bio- and CCU-based feedstocks as options for producing primary chemicals. Fossil-based feedstock: Primarily includes natural gas (i.e., methane and ethane), oil (i.e., LPG, naphtha and heavy oil), and coal. Natural gas can be produced in fields associated with oil or in dedicated fields. Ethane and methane can be recovered from natural gas and used for steam cracking and steam methane reforming units, respectively. LPG, naphtha and heavy oil are outputs of the petroleum refining sector, which is described in ref.¹².

LPG is an input to propane dehydrogenation process, naphtha can be used in steam cracking and catalytic reforming units whereas heavy oil is used for hydrogen to ammonia production via partial oil oxidation. Coal is produced in different qualities in the resource module (bituminous, subbituminous, and lignite) but only bituminous coal is allowed in coal gasification processes.

Bio-based feedstock: Primarily includes solid and liquid biofuel produced from planted forests and energy crops. Solid biofuel is obtained from woody (eucalyptus) and grassy sources as well as agriculture residues, including sugarcane bagasse. Eucalyptus can be used for biomass gasification processes to produce hydrogen (for energy or ammonia production), methanol, or can be used in Fischer-Tropsch Biomass-to-Liquids (FT-BtL) to produce advanced liquid biofuels. Sugarcane bagasse can be used for 2G bioethanol production.

Liquid biofuels included as feedstocks for the chemical sector are bioethanol (via fermentation/distillation from sugarcane, maize, wheat, sugar beet, and agricultural residues), and bio-naphtha, and bio-LPG as co-products of hydrotreated vegetable oils (HVO) or hydroprocessed esters and fatty acids (HEFA) production from soybean oil, maize oil, or animal fat. HVO and HEFA are advanced biofuels alternatives to be used in diesel engines as green diesel or jet fuel (or sustainable aviation fuel – SAF), and might co-produce bio-naphtha and bio-LPG to be used in other sectors, including as feedstocks.

Biomethane could be a promising alternative feedstock. However, its supply chain is not yet developed in COFFEE and it remains as a topic to be further explored in future work.

CCU-based feedstock: CO₂ as a feedstock is sourced from a “CO₂ pool”, which is composed by CO₂ captured from: 1) fossil- or bio-based concentrated sources such as power plants, cement kilns, blast furnaces, and methanol/ammonia plants (via gasification/partial oxidation, but not electrolysis). Both energy and process emissions can be captured from concentrated sources, and costs for CO₂ transportation are considered; 2) diluted CO₂ in the atmosphere via Direct Air Capture (DAC) based on absorption in sodium hydroxide solution, based on refs,^{13,14}. Captured CO₂ feeds into a regional “CO₂ pool” in the model, where the model decides whether it is geologically stored or used for CCU.

S2.4. Final disposal assumptions

Solid waste management data from World Bank’s What a Waste 2.0 database was used as a baseline to estimate regional shares for incineration, landfilling (i.e., controlled landfilling plus mismanagement) and recycling (see Figure S4). Incineration and recycling emissions were calculated based on ref.85. For plastics incineration, it was assumed that the carbon content of each HVC was emitted as CO₂. Therefore, 3.14 tCO₂/t ethylene, 3.14 tCO₂/t propylene, 3.25 tCO₂/t butadiene, and 3.38 tCO₂/t aromatics (using benzene as a proxy) were assumed when accounting for those emissions. Recycling considers 10% of material loss at every cycle and emissions account for electricity use from shredding,

extrusion, and agglomeration.

Figure S4. Final disposal shares assumed in the base year for COFFEE regions. Source: own elaboration based on ref. 86.

We also added a sentence in the Results section/Emission pathways for the global chemical sector:

While accounting for biogenic carbon storage in bioplastics highlights the potential of carbon removal in long-lived application as a mitigation strategy, it is essential to assess other sustainability dimensions to fully capture its effectiveness as a carbon mitigation strategy.

Comment 8	Lines 260-276 Please elaborate more carefully under which conditions and from what processes the chemicals sector and specifically hydrogen production becomes net negative.
Response	We thank the reviewer for the comment. Here, we consider only direct emissions and removals (Scope 1) from the chemical sector, which includes the explicitly represented processes (primary chemicals) and other chemicals (inorganic chemicals, implicitly modelled following IEA’s energy use and assuming historical growth rates). Emissions related to hydrogen production as feedstock are accounted as process emissions; emissions related to hydrogen production as energy carrier are accounted as energy emissions. The chemical sector becomes net negative mainly via scaling up biomass gasification coupled with CCS to produce hydrogen for ammonia production. We also account for biogenic carbon storage in plastics, which makes some contributions as well.
Notes/actions	We rewrote this paragraph as follows: FROM: Results on syngas products show that biomass gasification is prioritized for ammonia production due to the potential for BECCS, whereas CCU is prioritized for methanol production (Error! Reference source not found. d, e). This happens because, in ammonia production, hydrogen is the primary product, leading to either negative emissions with biogenic sources or emission reductions with fossil sources, depending on the extent of carbon capture. However, water electrolysis coupled with O2 use to

capture CO₂ via oxyfuel routes might also be promising. For methanol production, part of the carbon content remains in the material. Overall, the use of fossil fuels for ammonia reduces over time but it remains relevant, especially in more restrictive scenarios.

Emission pathways for the global chemical sector: from climate burden to asset

Figure 4 shows the role of the chemical sector in a 1.5°C world subject to different restrictions. Without any constraints on biomass use, CCS scale-up, and considering that biomaterials store biogenic carbon, the chemical sector reaches the levels of -746 MtCO₂.yr⁻¹ by 2050 (scenario 1.5C). Not only the chemical sector does not always behave as hard-to-abate from an emissions standpoint, but it can also be a climate asset. Emissions reduction stems mostly from BECCS in ammonia production and biogenic carbon storage, as well as increased efficiency in the fossil fuels-based platforms. However, when all constraints are turned on in the model, the chemical sector remains with residual emissions in 2050, and reaches net-negative emissions only by 2070 (scenario all). This difference of ~1Gt CO₂.yr⁻¹ increases the burden to other sectors, that need to extend 280 and anticipate decarbonisation efforts.

TO: Results on syngas products show that biomass gasification with CCS is favoured for ammonia production due to the potential for BECCS whereas CCU is preferred for methanol production (Figure 3 d, e). While in ammonia production the carbon is largely converted into captured CO₂ with high purity, contingent on the efficiency of the capture process, in methanol production, some of the carbon remains in the final product. Therefore, scaling up biomass use with carbon capture in ammonia production is identified by the model as a source of negative emissions. This can significantly reduce emissions across the overall system both in scale and pace required for reaching 1.5-degrees goals. Nevertheless, water electrolysis coupled with O₂ use in oxyfuel routes for capturing CO₂ could also be effective, although not assessed in this work. Overall, the use of fossil fuels for ammonia reduces over time but remains relevant, especially in more restrictive scenarios.

Emission pathways for the global chemical sector: from climate burden to asset

Figure 4 shows the role of the chemical sector in a 1.5°C world subject to different restrictions. It achieves a net reduction of -0.73 GtCO₂.yr⁻¹ in scenario 1.5C and -1GtCO₂.yr⁻¹ in scenario gCCS by 2050. Direct emissions reduction stems mostly from BECCS in ammonia production and biogenic carbon storage, as well as increased efficiency in fossil fuels-based platforms. Not only does the chemical sector not always behave as hard-to-abate from an emissions standpoint, but it can also be a climate asset if bio-based resources are available and the potential of storing biogenic carbon in biomaterials is explored. However, when all constraints are turned on in the model, the chemical sector remains with residual emissions in 2050 and reaches net-negative emissions only by 2070 (scenario all). This difference of ~1Gt CO₂.yr⁻¹ increases the burden on other sectors, which in turn have to extend and anticipate decarbonisation efforts.

Comment 9	Line 312-325 These are very interesting results that should be drawn out. Also, there are lots of English errors, see above.
Response	Thanks for your comment. We improved the writing and content of the discussion section, including this passage, as detailed below.
Notes/actions	FROM: On the chemical industry structure, steam cracking remains the main technology for HVCs production over the century. This is largely influenced by electrification of the transport sector, which results in more gasoline being diverted to the naphtha pool. However, refinery sourced HVCs reduce participation due to the shrinking of the refining sector and on purpose routes increase participation to accommodate feedstock substitution, particularly routes based on methanol (via MTO, MTA) and ethanol (via BDH) as intermediates. Meanwhile, the methanol and ammonia production undergoes a complete restructuring. While fossil fuels remain the main feedstock to HVCs, biomass gasification with CCS becomes the winning route for ammonia, whereas CCU is seen as key for methanol production. Moreover, methanol increases demand up to 3 times the levels used for non-energy purposes today due to its relative relevance as HVC intermediate. This indicates that solutions are sector-specific and all carbon feedstock sources will be relevant to achieve climate targets. Nonetheless, the relevance of fossil fuel use for non-energy purposes amplifies with CCS and biomass availability restrictions. TO: This work contributes to the research field of carbon emissions mitigation in the primary chemicals sector in four ways. Firstly, we observe that primary chemicals are hard-to-abate but can have a role in promoting systemic decarbonisation. Although decarbonisation technologies are available and could reduce emissions to as low as -1 GtCO₂yr⁻¹ by 2050, these reductions heavily rely on the large-scale availability of alternative feedstocks and on accounting for bio-based carbon storage in bioplastics that are neither incinerated nor recycled. In scenarios where alternative feedstocks are limited and carbon storage in biomaterials is not assumed, achieving net-zero emissions in the chemicals sector by 2070 remains elusive. Secondly, primary chemicals are hard-to-defossilize, leading to critical implications for decarbonisation of the chemicals sector and beyond. Our findings suggest that ambitious feedstock substitution could reduce fossil fuel dependence by approximately 60% by 2050 globally. However, if carbon capture and biomass use are constrained, the feedstock use profile remains essentially unchanged. This reflects both an opportunity and a challenge. On the one hand, substituting feedstock can significantly reduce refinery utilisation bringing indirect decarbonisation across sectors. On the other hand, failing to transition away from fossil fuels use as feedstock could inadvertently sustain petroleum refining activities, prolonging fossil fuel reliance and delaying the broader transition away from fossil fuels. This scenario could lead to continued fossil fuel extraction and refining focused on feedstock production, potentially resulting in lower-cost fuel co-production, and assets becoming stranded before investments are amortised. These findings highlight critical implications for investments in fossil fuel assets and underscore the need for a more integrated approach to energy, climate, and resource regulation. Thirdly, transitioning away from fossil fuels will require restructuring within the primary chemicals sector. Steam cracking remained the leading

	technology for HVC production over the century, largely influenced by the electrification of the transport sector and the resulting diversion of gasoline to the naphtha pool. However, refinery-sourced HVCs will decline due to the shrinking of the refining sector as a result of fossil fuels phase-out policies. On-purpose routes, particularly the ones based on methanol (via MTO, MTA) and ethanol (via BDH, and also influenced by oversupply due to transport electrification) as intermediates, will increase to accommodate feedstock substitutions. The restructuring will also affect methanol and ammonia production; biomass gasification with CCS becomes the preferred route for ammonia, while CCU plays a key role in methanol production. Demand for non-energy methanol in mitigation scenarios is expected to increase two to five times by 2050 compared to NPi. This is primarily due to its importance as an intermediate for HVC production. This shift indicates that solutions are product-specific and every carbon feedstock source will be relevant for achieving climate targets. Importantly, our scenarios explore a limited set of technologies to understand the role of feedstock supply in achieving net-zero scenarios, assuming unchanged demand patterns for primary chemicals. Given the diverse products and services provided by primary chemicals, detailing and addressing their complexities is challenging. This work aimed to unravel the dynamics of petroleum production and refining within the emissions pathway of primary chemicals by developing drop-in substitution alternatives. Nevertheless, we acknowledge that more is needed to provide a comprehensive view of the potential futures lying ahead for the chemical sector. Historically, energy transitions have redefined systems and triggered material transitions and vice-versa³⁶. Hence, transforming the complex network of feedstocks, products, and services developed under the petroleum-based economy and thermochemical foundations since the 1950s will require innovative solutions. Beyond phasing-out fossil fuels in energy systems, these include the development of biodegradable chemicals, material substitutions, and demand-side measures. While these aspects were not the focus of our assessment, they are crucial considerations for future policymaking. Fourth and lastly, improving the representation of fossil fuels in IAMs to capture relevant material-energy links allows modellers to reach a higher degree of accuracy in depicting systems integration. Representing incumbent energy carriers set to be phased-out in detail enables modellers to provide realistic but ambitious recommendations to policymakers considering their pervasive use across sectors. In this sense, the scope and scale of the required transition in the chemical sector to respond to the climate challenges of the 21st century are diverse and in different stages of the supply chain.
--	---

Comment 10	Line 464 The first caveat on not including demand and end of life should come much earlier in the paper. It is critical.
Response	We thank the reviewer for highlighting this. We agree that this is a critical consideration that should be mentioned earlier to the readers.
Notes/actions	We included that information in two parts of the manuscript as follows. In the Main section. FROM: We explore global technological pathways, carbon feedstocks, energy use, and direct emissions scenarios for the basic chemicals

	industry in scenarios considering Implemented National Policies (NPI) and climate policies aiming at limiting global average temperature increase to below 1.5oC above pre-industrial levels (1.5C) by 2100. TO: With a focus on supply-side mitigation measures, we explore global technological pathways, carbon feedstocks, energy use and direct emissions scenarios for the primary chemicals industry in scenarios considering Implemented National Policies (NPI) and climate policies aiming at limiting global average temperature increase to below 1.5oC above pre-industrial levels (1.5C) by 2100. In the Online Methods section: FROM: Scenario development All of the scenarios are built on the assumptions of the Shared Socioeconomic Pathway 2 – the “middle of the road” scenario, which extrapolates historical patterns of social, economic, and technological trends throughout the century ^{38,39}. The assumptions made for each scenario are described below and summarized in Table S4 in the SI. TO: Scenario development All of the scenarios are built on the assumptions of the Shared Socioeconomic Pathway 2 – the “middle of the road” scenario, which extrapolates historical patterns of social, economic, and technological trends throughout the century ^{38,39}. The assumptions made for each scenario are described below and summarized in Table S4 in the SI. Plastics end-of-life and demand patterns were assumed to remain the same throughout the century to allow a more detailed analysis of the supply-side.
--	---

Comment 11	While I wasn't able to check as I went through because of the “error references...” etc, I am surprised there is no referencing to the IPCC AR6 Industry chapter (Bashmakov et al., 2022)/, specifically sections 11.3 and 11.4. It discussed decarbonisation pathways for chemicals in some detail based on the bottom up literature for the sector.
Response	We thank the reviewer for pointing this out. While it was not explicitly mentioned to back any specific sentence in the text, we consider the IPCC reports, including the AR6 Industry chapter, as key references in our discussions and analysis.
Notes/actions	We referenced to the IPCC AR6 Industry chapter in a sentence in the Main section (reference 22): Despite the decreasing demand for fossil fuels in energy systems due to climate policies^{3,20,21}, the co-production of energy, feedstocks, and chemicals in petroleum refineries and implications to energy use and emissions pathways across sectors remains largely unexplored²².

Comment 12	Despite the grammatic issues, We thank the reviewer for an easy to read & interesting paper, and I hope the journal continues with it. References
-------------------	---

	Bashmakov, I., Nilsson, L., Acquaye, A., Bataille, C., Cullen, M., de la Rue du Can, S., Fishedick, M., Geng, Y., Tanaka, K., Bauer, F., Hasanbeigi, A., Levi, P., Myshak, A., Perczyk, D., Philibert, C., & Samadi, S. (2022). Chapter 11: Industry. IPCC AR6 WGIII Mitigation. IPCC. https://www.ipcc.ch/report/ar6/wg3/ Hepburn, C., Adlen, E., Beddington, J., Carter, E. A., Fuss, S., Mac Dowell, N., Minx, J. C., Smith, P., & Williams, C. K. (2019). The technological and economic prospects for CO2 utilisation and removal. Nature, 575(7781), 87–97. https://doi.org/10.1038/s41586-019-1681-6 Tanzer, S. E., & Ramírez, A. (2019). When are negative emissions negative emissions? Energy and Environmental Science, 12(4), 1210–1218. https://doi.org/10.1039/c8ee03338b
Response	We appreciate the feedback and are pleased to hear the reviewer found the paper engaging. We look forward to our continued collaboration with the journal.
Notes/actions	N/A

REVIEWER COMMENTS

Reviewer #1 (Remarks to the Author):

Many thanks to the authors for their hard work. The revised manuscript has indeed been much improved. However, there are still several issues, which need to be addressed before acceptance for publication.

Title: Unaddressed non-energy use in the chemical industry can undermine fossil fuels phase out

Comment: It sounds like we still need to phase out fossil fuels thus we need to pay attention to its non-energy use in the chemical industry. However, after reading the manuscript, my take-away is it's fine not to phase out fossil fuels. We can still reach net-zero. Is that your key message?

P6: We argue that the interdependent relationship between fuels and chemicals means that efforts in the chemicals sector are relevant not only to achieving sectoral net-zero but also to reducing the utilisation factor of petroleum

refineries. This leads to indirect lower fossil fuel use and mitigation across all sectors. Moreover, as alternative routes for chemicals rely on the supply of biomass, H₂ and CO₂, results from an integrated assessment perspective should reveal potential competition for these resources between energy and non-energy uses and between carbon capture and utilisation (CCU) and CCS.

Comment: I don't understand this paragraph added as research objective.

P20: Therefore, scaling-up biomass use with carbon capture in ammonia production is identified by the model as a source of negative emissions. Nevertheless, water electrolysis coupled with O₂ use in oxyfuel routes for capturing CO₂ could also be effective, although not assessed in this work.

Comment: Is electrified Haber-Bosch for ammonia production in the model? Even it's true that biomass gasification with CCS is the preferred solution, we can argue why not produce more methanol and increase MTO/MTA routes, rather than wasting all the carbon in the biomass to produce hydrogen.

P23: Fig 4: 'Avoided emissions' reflects the reduction in emissions achieved through the CCS of fossil feedstock used in the production to produce ammonia and methanol.

Comments: I don't understand this part. And if it's avoided emissions, why it is positive in the figure? Also, explain "Bio ICCS" and "MNET" in the legend.

In the gCCS scenario, why more CO₂ is captured for storage than other scenarios, if the scenario is set to constraint global CCS deployment?

P23: These reductions heavily rely on the large-scale availability of alternative feedstocks and on accounting for bio-based carbon storage in bioplastics that are neither incinerated nor recycled

Comments: How is recycling considered in the study? Recycling decreases the demand for virgin HVC production, or increases supply of carbon through pyrolysis or gasification

Comment 13, and many claims in the text regarding the climate benefits of plastics:

Comments: I'm still not convinced with this high negative emission number, especially regarding the EoL treatment assumption that it would stay the same as today (mostly landfilling).

Reviewer #2 (Remarks to the Author):

The revised manuscript has been significantly improved. I still have two comments as below.

1) The revised sentence below in the abstract still looks strange. I am wondering how I should interpret the phrase "could be substituted". Is it maximum percentage?

"Using a global integrated assessment model with detailed refining and primary chemicals sectors, findings reveal that up to 62% of fossil fuels used as feedstock in the chemical sector could be substituted by alternatives by 2050 with significant scale-up of biomass utilisation and carbon capture technologies."

2) Regarding the response shown below, as I previously pointed out, the hydrogen generation cost is related to how the electricity dispatch is considered. In the earlier studies, the hydrogen would be normally generated by the VRE since battery or water electrolysis are the major technologies to adjust the variability of the renewable energy. The way how those dispatch is managed is directly hit electricity generation cost changing hydrogen cost. Eventually they would influence on the cost of alternative fuels. Is it really appropriate treatment for this study? authors have not yet been convinced.

“We agree with the reviewer on the relevance of DAC and hydrogen technologies as well as their costs given how the renewable power is modelled. Although hydrogen techno economic assumptions are described in the SI, DAC was not fully described in the SI and Online methods because it is a carbon removal technology that is not represented within the chemical sector. DAC has a link with the chemical sector via a “CO2 pool”. We clarify this in the bullet points below:

1. Hydrogen can be produced via various routes. Water electrolysis (using both renewable and non-renewable electricity), biomass gasification (w/ and w/o CCS), coal gasification (w/ and w/o CCS), partial oxidation of oil and steam methane reforming (w/ and w/o CCS). It is also co-produced in steam cracking processes and catalytic reform of naphtha. COFFEE does not apply constraints in electricity generation. Thus, the model can deploy conventional or alternative electricity generation technologies to 1) Fulfil electricity demand in the current level of participation, 2) Electrify further energy services and 3) Produce hydrogen or store electricity in batteries.”

Reviewer #3 (Remarks to the Author):

To the authors -

Thank you for an easy to review paper.

Summary

- The paper is much better than the first pass.
- There are few issues still, see below.
- Summary – minor revision. My final assessment is that this paper has something to contribute to the literature, but still needs a bit of work.

Editorial

- “becomes” is extraneous in the abstract

Substantive

- Page 8 – I think you may have double negated yourself here “Global CO₂ emissions in the scenario with all restrictions must be reduced by at least 6 GtCO₂ per year less than in scenario 1.5C by 2030 to stay within the CO₂ budget.” Shouldn’t “less” be “more”?
- Page 9 “The resurgence in coal use observed after mid-century aligns with the literature and is attributed to the combining effects of: 1) a rising energy demand in regions projected to experience significant population and affluence growth post-2050 (i.e., Africa, Southeast Asia, and India), 2) escalating costs of oil and gas, and 3) coal use transitioning to include carbon capture 3.” I am somewhat suspicious of point 3. No one anywhere has managed to achieve robust, reliable 90%+ capture on coal plants – for electricity, for steel, for cement – one reference does not a robust literature make. I would soften your surety here somewhat.
- Page 13 – In the response to reviewers (reviewer 3#) you mention that total transport demand of all types maxes out at 37%. This is not going to be intuitive to readers that have gotten used to the idea that personal transport demand will largely electrify. You need to find some way (text, figure) to transparently show the level of liquids to electricity displacement for personal and freight ground road transport, rail, shipping and aviation fuel. Some may disagree with your results, but they will be much clearer and transparent.
- Reviewer 3#, comment 7 – You haven’t addressed one of the critical points of the reviewer, about the assumption of biomass/energy carbon neutrality. You have noted that COFFEE includes land use emissions, but you stated what you assume about harvest species, harvest method, soil carbon effects, etc. All these things could lead to biomass or bioenergy being very far from carbon neutral. Just state your assumption of neutrality and the basis for it.

Point-by-point response to reviewers

2nd round

Dear Editor and Reviewers,

We wish to express our appreciation for your in-depth comments, suggestions, and corrections, which have greatly improved the manuscript's quality and depth. In this letter, we respond to each point raised in the 2nd round of reviews, clearly indicating the amendments made to our updated manuscript *Unaddressed non-energy use in the chemical industry can undermine fossil fuels phase-out*.

Once again, and for the sake of clarity:

- i) Reviewers' comments are written in plain text;
- ii) Our responses are in blue;
- iii) The text quoted from the paper is green.

Reviewer #1:

Many thanks to the authors for their hard work. The revised manuscript has indeed been much improved. However, there are still several issues which need to be addressed before acceptance for publication.

We thank the reviewer for the encouraging feedback and for acknowledging the improvements made to our manuscript. We appreciate the thorough review and constructive comments.

Comment 1	It sounds like we still need to phase out fossil fuels thus we need to pay attention to its non-energy use in the chemical industry. However, after reading the manuscript, my take-away is it's fine not to phase out fossil fuels. We can still reach net-zero. Is that your key message?
Response	Thank you for your insightful comment. Our key message is not that it is fine not to phase out fossil fuels entirely. The scenario results indicate lower but continued use of fossil fuels post-net-zero for energy and non-energy purposes, primarily oil and gas, with some coal use equipped with CCS. This outcome means that the COFFEE model found a cost-optimal solution for net-zero pathways with a carbon budget consistent with limiting global warming to 1.5°C, given the constraints applied and the population and economic projections assumed (SSP2). However, translating these model results into climate policy and action involves significant technical (e.g., scaling up biomass, CCS, hydrogen, DAC, along with chemical recycling, changes in design, and other circular economy approaches), political (e.g., policy implementation, international agreements) and societal (e.g., public acceptance, behavioural changes) challenges. It also involves uncertainties such as the actual population and GDP growth and distribution, wars, pandemics, and climate impacts, to name a few. Achieving the necessary scale of effort indicated in these results, especially within this decade, is not impossible but extremely challenging; the burden of not phasing-out fossil fuels is already too significant. However, we recognize the limitations of the options included in our study. We did not represent demand-side measures, chemical recycling, COTC, biodegradable plastics, lifetime extension, and others not because there is

	no scientific evidence that they could be relevant, but because the representation of these alternatives is limited by our modelling framework or data availability. While key results like biomass use in ammonia or plastics used in long-lived applications point to potential opportunities for the chemicals sector, IAMs are only one of the many available modelling tools that could and should be used to support decision-making. Beyond GHG emissions, other socio-environmental perspectives should be considered in environmental policymaking. Therefore, the main takeaways should not be the absolute numbers found in these cost-optimal trajectories but understanding how systems interact. Specifically, our research emphasizes that:  1) The chemical sector is hard-to-abate but can develop strategies to ease the burden in other sectors if the right policies and conditions are put in place; 2) Reducing oil use for non-energy purposes has direct and indirect mitigation impacts, which are not usually accounted; 3) Decarbonization of chemicals requires a complete restructuring of the industry as we currently know; 4) Mitigation in various sectors and feedstock substitution compete for the same resources (biomass, CCUS) and should be managed carefully under CCS/biomass-limited scenarios; and 5) Fossil fuels phase-out policies should go hand-in-hand with the chemicals sector decarbonization. These strategies are interdependent and have mutual impacts. We hope this clarifies our position and the implications of our findings.
Notes/actions	N/A

Comment 2	P6: We argue that the interdependent relationship between fuels and chemicals means that efforts in the chemicals sector are relevant not only to achieving sectoral net-zero but also to reducing the utilisation factor of petroleum refineries. This leads to indirect lower fossil fuel use and mitigation across all sectors. Moreover, as alternative routes for chemicals rely on the supply of biomass, H2 and CO2, results from an integrated assessment perspective should reveal potential competition for these resources between energy and non-energy uses and between carbon capture and utilisation (CCU) and CCS. Comment: I don't understand this paragraph added as research objective.
Response	Thank you for your comment. With this paragraph, we aim to link our research objective to the scenarios designed for this study. In our view, there are two key features of the chemical sector that deserve a better assessment:  ■ Integration with the oil refining sector: fuels and chemicals are co-produced in oil refining plants with limited flexibility; phasing-out fossil fuels in energy systems seems out of sight if demand for fossil fuels as feedstock is increasing. ■ Resource allocation/competition: As heavily relying on biomass use and CO₂ (from CCS) for carbon feedstock, feedstock substitution directly competes with other mitigation measures across sectors that also rely on them to reduce emissions.

	We made the following changes to improve the clarity of our text, and we hope that they address your concerns.
Notes/actions	FROM: Despite the decreasing demand for fossil fuels in energy systems due to climate policies, the co-production of energy, feedstocks, and chemicals in petroleum refineries and implications to energy use and emissions pathways across sectors remain largely unexplored. Aiming to fill this gap, this article investigates the role of the chemical sector in a global net-zero strategy. Our hypothesis is that the chemical sector, while being both hard-to-abate and hard-to-defossilize, can offer strategic contributions to deep decarbonisation, both sectoral and systemically. We argue that the interdependent relationship between fuels and chemicals means that efforts in the chemicals sector are relevant not only to achieving sectoral net-zero but also to reducing the utilisation factor of petroleum refineries. This leads to indirect lower fossil fuel use and mitigation across all sectors. Moreover, as alternative routes for chemicals rely on the supply of biomass, H₂ and CO₂, results from an integrated assessment perspective should reveal potential competition for these resources between energy and non-energy uses and between carbon capture and utilisation (CCU) and CCS. TO: Despite the decreasing demand for fossil fuels in energy systems due to climate policies, the co-production of energy, feedstocks and chemicals in petroleum refineries, and implications to energy use and emissions pathways across sectors remain largely unexplored. Aiming to fill this gap, this article investigates the role of the chemical sector in a global net-zero strategy. Our hypothesis is that the chemical sector, while being both hard-to-abate and hard-to-defossilize, can offer strategic contributions to deep decarbonisation, both sectoral and systemically. An integrated assessment perspective is essential to understand: 1) How the integration of a growing chemical sector with the oil refining sector affects fossil fuels phase-out and decarbonization of chemicals; and 2) The competition for CO₂ (CCUS), hydrogen, and biomass for feedstock substitution with other mitigation measures across sectors that also rely on these resources to reduce emissions.

Comment 3	P20: Therefore, scaling-up biomass use with carbon capture in ammonia production is identified by the model as a source of negative emissions. Nevertheless, water electrolysis coupled with O₂ use in oxyfuel routes for capturing CO₂ could also be effective, although not assessed in this work. Comment: Is electrified Haber-Bosch for ammonia production in the model? Even it's true that biomass gasification with CCS is the preferred solution, we can argue why not produce more methanol and increase MTO/MTA routes, rather than wasting all the carbon in the biomass to produce hydrogen.
Response	Thank you for your comment. Electrified Haber-Bosch is included in the model through 'Electrolysis' in H₂ production (Figure3-d). From a sectoral net-zero perspective, your logic is understandable. However, the main advantage of an integrated assessment perspective is that it helps identify the optimal solutions for the whole system and not just for individual sectors. The model allocates biomass and CCS in ammonia production because a large share of the carbon goes to long-term storage (all the carbon except what is consumed for urea production + losses), leading to net negative carbon emissions. This decision is influenced by the ambitious decarbonization target, which also favors hydrogen production

	via biomass gasification with CCS. If the carbon were used for methanol production and then through MTO/MTA routes, there would be more conversion steps involved, leading to more efficiency losses and a need to remove carbon elsewhere (or significantly changing the rest of the emission profile of the economy to reduce the need for negative emissions). Therefore, as a linear programming model, the solution in COFFEE tends to the cost-optimal scenario, favoring the use of biomass and CCS for ammonia production.
Notes/actions	N/A

Comment 4	P23: Fig 4: ‘Avoided emissions’ reflects the reduction in emissions achieved through the CCS of fossil feedstock used in the production to produce ammonia and methanol. Comments: I don’t understand this part. And if it’s avoided emissions, why it is positive in the figure? Also, explain “Bio ICCS” and “MNET” in the legend. In the gCCS scenario, why more CO2 is captured for storage than other scenarios, if the scenario is set to constraint global CCS deployment?
Response	Thank you for your questions and suggestion. “Avoided emissions” are represented as positive avoided emissions, as opposed to carbon removal options, which can actually lead to removals from the atmosphere and are represented as negative emissions. They are, essentially, an overall reduction in the positive emissions. In contrast, BECCS, MNET, and Bio-ICCS refer to actual net removals. Furthermore, in the gCCS scenario, we see is that the chemical sector is prioritized for capture across various sectors (see Supplementary Fig. 15). Although the global CCS roll-out declines in 2050 compared to other scenarios, the use of CCS in the chemical sector increases because the model chooses to reduce emissions in the sector with the lowest cost. Note that when biomass and CCS are constrained simultaneously (scenario all), this relative advantage disappears. Similarly, when only biomass is constrained, total CCS remains the same but “Energy CCS – Fossil fuels” increases in relative terms, leading to a greater relevance of

'DAC'

We hope we have clarified your questions and that the actions taken below address your concerns.

Notes/actions

The emissions figure was replaced by a new one with hatched bars indicating CCS. Now, avoided emissions are described as 'Process CCS (Fossil)' and Bio CCS as 'Process CCS (Bio)', as below. We also applied hatch patterns to the bars to distinguish CCS from emissions (positive) and removals (negative):

We also updated the figure caption accordingly:
FROM: Figure 4. Global direct CO₂ emissions from the chemical sector. Includes direct (Scope 1) emissions from primary chemicals (explicitly modelled in COFFEE) and other chemicals (implicitly modelled in

COFFEE) (see the Glossary and definitions section in the Supplementary Information file). Symbols represent net CO₂ emission in each scenario. Process and Energy emissions indicate residual CO₂ emissions resulting from fossil fuel use either as an energy source or as a feedstock in chemical production. Emissions from incineration and recycling are also allocated as Process Emissions. 'Avoided emissions' reflects the reduction in emissions achieved through the CCS of fossil feedstock used to produce ammonia and methanol. Supplementary Figure 12 presents regional pathways for CO₂ emissions in the chemical sector across R5 regions.

TO: Figure 4. Global direct CO₂ emissions from the chemical sector. Includes direct (Scope 1) emissions from primary chemicals (explicitly modelled in COFFEE) and other chemicals (implicitly modelled in COFFEE) (see the Glossary and definitions section in the Supplementary Information file). Symbols represent net CO₂ emissions in each scenario. Energy and Process emissions refer to CO₂ emissions resulting from combustion and from chemical reactions not related to energy use, respectively. In this work, we consider emissions from hydrogen production for ammonia synthesis as process emissions. Emissions from incineration and mechanical recycling are also regarded as Process Emissions. 'Process CCS (Fossil)' reflects the reduction in emissions achieved through the CCS of fossil feedstock used to produce ammonia and methanol. 'Process CCS (Bio)' refers to the removal of CO₂ emissions achieved through the CCS of bio-based feedstock used to produce ammonia and methanol. Supplementary Figure 12 presents regional pathways for CO₂ emissions in the chemical sector across R5 regions.

We also included a paragraph in the results section and a Supplementary Figure to support the point discussed above about gCCS:

Furthermore, in the gCCS scenario, we see is that the chemical sector is prioritized for capture across various sectors (see Supplementary Fig. 15). Although the global CCS roll-out declines in 2050 compared to other scenarios, the use of CCS in the chemical sector increases because the model chooses to reduce emissions in the sector with the lowest cost. Note

	that when biomass and CCS are constrained simultaneously (scenario all), this relative advantage disappears. Similarly, when only biomass is constrained, total CCS remains the same but “Energy CCS – Fossil fuels” increases in relative terms, leading to a greater relevance of ‘DAC’.
--	--

Comment 5	P23: These reductions heavily rely on the large-scale availability of alternative feedstocks and on accounting for bio-based carbon storage in bioplastics that are neither incinerated nor recycled Comments: How is recycling considered in the study? Recycling decreases the demand for virgin HVC production, or increases supply of carbon through pyrolysis or gasification
Response	Thank you for your comment. Recycling decreases the demand for virgin HVC as we consider only mechanical recycling. Therefore, energy use for shredding, extrusion, and agglomeration is also accounted for. Chemical recycling (pyrolysis, gasification, or solvolysis) is out of our scope, but we agree that this should have been mentioned as a relevant direction for future studies.
Notes/actions	FROM: Apart from the limitations mentioned before, a significant albeit deliberate limitation was not including the demand-side or final disposal insights. It is indisputable that these measures hold a highly relevant potential. Stegmann (2022), for instance, found that recycled plastics could make up to 60% of plastics produced yearly by 2050. TO: Apart from the limitations mentioned before, a significant albeit deliberate limitation was not including the demand-side or final disposal insights. It is indisputable that these measures hold a highly relevant potential. Stegmann (2022), for instance, found that recycled plastics could make up to 60% of plastics produced yearly by 2050. Chemical recycling (e.g., via pyrolysis, gasification, solvolysis) is also highlighted as an alternative to drastically reduce the future demand for virgin plastics (Coates & Getzler, 2020; Thiyagarajan et al., 2022). FROM: S2.4. Final disposal assumptions Solid waste management data from World Bank’s What a Waste 2.0 database was used as a baseline to estimate regional shares for incineration, landfilling (i.e., controlled landfilling plus mismanagement) and recycling (see Supplementary Figure 4). Incineration and recycling emissions were calculated based on ref.85. For plastics incineration, it was assumed that the carbon content of each HVC was emitted as CO₂. Therefore, 3.14 tCO₂/t ethylene, 3.14 tCO₂/t propylene, 3.25 tCO₂/t butadiene, and 3.38 tCO₂/t aromatics (using benzene as a proxy) were assumed when accounting for those emissions. Recycling considers 10% of material loss at every cycle and emissions account for electricity use from shredding, extrusion, and agglomeration. TO: S2.4. Final disposal assumptions Solid waste management data from the World Bank’s What a Waste 2.0 database were used as a baseline to estimate regional shares for incineration, landfilling (i.e., controlled landfilling plus mismanagement) and mechanical recycling (see Supplementary Figure 4). Incineration and mechanical recycling emissions were calculated based on ref.85. For plastics incineration, it was assumed that the carbon content of each HVC was emitted as CO₂. Therefore, 3.14 tCO₂/t ethylene, 3.14 tCO₂/t

	propylene, 3.25 tCO₂/t butadiene, and 3.38 tCO₂/t aromatics (using benzene as a proxy) were assumed when accounting for those emissions. Mechanical recycling considers 10% of material loss at every cycle and emissions account for electricity use from shredding, extrusion, and agglomeration. FROM: Emissions from incineration and recycling are also allocated as Process Emissions. TO: Emissions from incineration and mechanical recycling are also allocated as Process Emissions.
--	---

Comment 6	Comment 13, and many claims in the text regarding the climate benefits of plastics: Comments: I'm still not convinced with this high negative emission number, especially regarding the EoL treatment assumption that it would stay the same as today (mostly landfilling).
Response	Thanks for your comment. The emissions mentioned refer to the whole chemical sector; therefore, emissions from the production of ammonia, methanol, HVCs, and other chemicals. Thus, we list below the reasons for the low emissions:  - Phase-out of coal in the chemical industry in 2030 in ammonia and methanol production. This dramatically reduces emissions in 2030 in mitigation scenarios. After 2050 there is a slight resurgence of coal but by then their emissions are compensated by the switch to biomass gasification + CCS (ammonia) and CCU (methanol). - Extensive capacity of Biomass gasification equipped with Carbon Capture for ammonia around 2050. This leads to the largest part of negative emissions in 2050 (-858 MtCO₂/yr in scenario 1.5C). - Methanol is largely supplied via CCU and Biomass gasification + CCS in 2050. In biomass- and CCS-constrained scenarios, CO₂ for CCU is largely supplied by Direct Air Capture. Fossil fuels are almost completely phased out in all scenarios. - Biogenic carbon storage in non-recycled and non-incinerated bio-based plastics by 2050. This is equivalent to -465 MtCO₂/yr in scenario 1.5C). We are not looking for now to the final disposal emissions, therefore this refers to plastics landfilled. This result indicates that if every bioplastic produced would be used in long-lived applications only (therefore diverted from landfilling), it would have relevant climate implications.

Notes/actions

N/A

Reviewer #2:

The revised manuscript has been significantly improved. I still have two comments as below.

Thank you for acknowledging the improvements in the revised manuscript. We appreciate your continued feedback, which has contributed to enhancing the quality of our work. Below are our responses to your two comments:

Comment 1	1) The revised sentence below in the abstract still looks strange. I am wondering how I should interpret the phrase “could be substituted”. Is it maximum percentage? “Using a global integrated assessment model with detailed refining and primary chemicals sectors, findings reveal that up to 62% of fossil fuels used as feedstock in the chemical sector could be substituted by alternatives by 2050 with significant scale-up of biomass utilisation and carbon capture technologies.”
Response	Thank you for your question. The percentage shows the maximum substitution share observed across the scenarios analysed. It does not indicate a maximum technical potential but rather reflects the results obtained within the scope and solution method of our model.
Notes/actions	The referred part of the abstract was rewritten to improve clarity. FROM: Using a global integrated assessment model with detailed refining and primary chemicals sectors, findings reveal that up to 62% of fossil fuels used as feedstock in the chemical sector could be substituted by alternatives by 2050 with significant scale-up of biomass utilisation and carbon capture technologies.

	TO: Using a global integrated assessment model with detailed refining and primary chemicals sectors, findings across various scenarios reveal that up to 62% of total feedstock use in the chemical sector could be provided by alternative sources by 2050. This would require significant scale-up in biomass utilisation and carbon capture technologies.
--	--

Comment 2	2) Regarding the response shown below, as I previously pointed out, the hydrogen generation cost is related to how the electricity dispatch is considered. In the earlier studies, the hydrogen would be normally generated by the VRE since battery or water electrolysis are the major technologies to adjust the variability of the renewable energy. The way how those dispatch is managed is directly hit electricity generation cost changing hydrogen cost. Eventually they would influence on the cost of alternative fuels. Is it really appropriate treatment for this study? authors have not yet been convinced. “We agree with the reviewer on the relevance of DAC and hydrogen technologies as well as their costs given how the renewable power is modelled. Although hydrogen techno economic assumptions are described in the SI, DAC was not fully described in the SI and Online methods because it is a carbon removal technology that is not represented within the chemical sector. DAC has a link with the chemical sector via a “CO2 pool”. We clarify this in the bullet points below: 1. Hydrogen can be produced via various routes. Water electrolysis (using both renewable and non-renewable electricity), biomass gasification (w/ and w/o CCS), coal gasification (w/ and w/o CCS), partial oxidation of oil and steam methane reforming (w/ and w/o CCS). It is also co-produced in steam cracking processes and catalytic reform of naphtha. COFFEE does not apply constraints in electricity generation. Thus, the model can deploy conventional or alternative electricity generation technologies to 1) Fulfil electricity demand in the current level of participation, 2) Electrify further energy services and 3) Produce hydrogen or store electricity in batteries.”
Response	Thank you for your comments. We appreciate the opportunity to clarify our results and address your concerns about how hydrogen is modelled in our study. We understand the role of dispatch modelling in the cost of hydrogen (hence, in alternative fuel costs), a subject that has been the topic of recent literature (Ueckerdt et al., 2015),. However, because of its long-term and global scope, global IAMs have limited representation of the spatiotemporal resolution required to model VRE variability. As far as we know, an explicit representation of VRE in global IAMs is lacking, although different methods exist to incorporate their features, such as soft-couplings with sectoral models with higher temporal and regional resolution (e.g., Gong et al.,2023). In the case of the COFFEE model, we implement regional power sector constraints that limit: 1) capacity expansion of power technologies, including VRE; 2) yearly VRE production in total electricity generation; 3) set minimum levels of "firm" electricity capacity options to compensate for intermittent capacity expansion. We also include utility-scale storage options (such as batteries), which allow the regional grid to increase the VRE share in generation (greater than the regional constraint). Another

strategy to bypass the regional constraint on VRE generation is using VRE-direct hydrogen production via electrolysis. These options consider different equipment utilization factors and storage as measures to deal with renewable variability and are more expensive than “conventional” electrolysis. Combining this strategy with the surplus VRE's low or zero shadow price, green hydrogen can become competitive in certain situations. Finally, hydrogen production via electrolysis can still be made from the electricity from the grid (what we called “conventional” previously), which uses a stable, dispatchable, and predictable source of electricity supply, leading to lower capital cost (balanced by the higher electricity cost). Therefore, the impact of VRE and storage options are somewhat represented in the model. Combined with all the other previously mentioned hydrogen production alternatives, we believe the hydrogen supply options are reasonably addressed and accounted for in our model. As mentioned before, this does not guarantee a detailed spatiotemporal resolution or replace a more detailed analysis of the matter, using more detail tools and methods.

Moreover, the cost of carbon weights more than the cost of hydrogen the more ambitious the scenario is. Additionally, it is important to highlight that the model contains different CO₂ capture options, including DAC, which compose a CO₂ pool. In scenarios where CCS and/or biomass are restricted, the DAC option gains importance (see Figure).

Likewise, in these restricted scenarios, methanol production relies heavily on the e-methanol route (Figure 3-e in the Manuscript). An important message here, then, is that in limited scenarios for the use of BECCS there are still ways (more expensive and with greater energy penalties) to carry out the technological transition of the chemical sector.

Gong, C. C., Ueckerdt, F., Pietzcker, R., Odenweller, A., Schill, W. P., Kittel, M., & Luderer, G. (2023). Bidirectional coupling of the long-term integrated assessment model REgional Model of INvestments and Development (REMIND) v3. 0.0 with the hourly power sector model Dispatch and Investment Evaluation Tool with Endogenous Renewables (DIETER) v1. 0.2. *Geoscientific Model Development*, 16(17), 4977-5033.

Notes/actions	N/A
---------------	-----

Reviewer #3:

Thank you for an easy to review paper.

Summary

- The paper is much better than the first pass.
- There are few issues still, see below.
- Summary – minor revision. My final assessment is that this paper has something to contribute to the literature, but still needs a bit of work.

Editorial

- “becomes” is extraneous in the abstract

Thank you for acknowledging the improvements of the revised version. Your feedback significantly contributed to enhancing its quality.

Below, we have revised the sentence in the abstract that was unclear, as noted.

FROM: Fossil fuel processing plants, such as petroleum refineries, exhibit interdependent material and energy system dynamics, making the transition away from fossil fuels energy systems becomes more challenging without addressing the non-energy outputs.

TO: Fossil fuel processing plants, such as petroleum refineries, exhibit interdependent material and energy system dynamics. This makes the transition away from fossil fuels in energy systems more challenging if non-energy outputs are not addressed.

Comment 1	Page 8 – I think you may have double negated yourself here “Global CO2 emissions in the scenario with all restrictions must be reduced by at least 6 GtCO2 per year less than in scenario 1.5C by 2030 to stay within the CO2 budget.” Shouldn’t “less” be “more”?
Response	You are right. Thank you for pointing that out.
Notes/actions	FROM: Global CO2 emissions in the scenario with all restrictions must be reduced by at least 6 GtCO2 per year than in scenario 1.5C by 2030 to stay within the CO2 budget. TO: Global CO₂ emissions in the scenario with all restrictions must be reduced by at least 6 GtCO₂ per year more than in scenario 1.5C by 2030 to stay within the CO₂ budget.

Comment 2	Page 9 “The resurgence(nce) in coal use observed after mid-century aligns with the literature and is attributed to the combining effects of: 1) a rising energy demand in regions projected to experience significant population and affluence growth post-2050 (i.e., Africa, Southeast Asia, and India), 2) escalating costs of oil and gas, and 3) coal use transitioning to include carbon capture 3.” I am somewhat suspicious of point 3. No one anywhere has managed to achieve robust, reliable 90%+ capture on coal plants – for
------------------	---

	electricity, for steel, for cement - one reference does not a robust literature make. I would soften your surety here somewhat.
Response	Thank you for your comment and suggestion. We agree with the argument above. However, we were referring to a paper that compiles hundreds of scenarios from the IPCC AR6 scenario database – and the three conclusions stated above stem from that. However, we agree that referring to the literature in general might be misleading when we are referring to a specific scenario literature and therefore we revised the paragraph.
Notes/actions	FROM: The resurgence in coal use observed after mid-century aligns with the literature and is attributed to the combining effects of: 1) a rising energy demand in regions projected to experience significant population and affluence growth post-2050 (i.e., Africa, Southeast Asia, and India), 2) escalating costs of oil and gas, and 3) coal use transitioning to include carbon capture 3.” TO: The resurgence in coal use observed after mid-century aligns with the IAM scenario literature and is attributed to the combining effects of: 1) A rising energy demand in regions projected to experience significant population and affluence growth post-2050 (i.e., Africa, Southeast Asia, and India), 2) Escalating costs of oil and gas, and 3) Coal use transitioning to include carbon capture.”

Comment 3	Page 13 – In the response to reviewers (reviewer 3#) you mention that total transport demand of all types maxes out at 37%. This is not going to be intuitive to readers that have gotten used to the idea that personal transport demand will largely electrify. You need to find some way (text, figure) to transparently show the level of liquids to electricity displacement for personal and freight ground road transport, rail, shipping and aviation fuel. Some may disagree with your results, but they will be much clearer and transparent.
Response	Thank you for the suggestion. We agree that the results might be a bit off the expectations of transportation electrification although freight transportation and aviation are hard-to-abate transportation modes precisely because they are not easily electrifiable and in some cases (long-distance sea and air transport) based on long-life converters, meaning that the fleet’s renewal is not immediate. As requested by the reviewer, we included the results for final energy use in freight (road and rail) and passenger transportation. Detailed shipping results can be found in Muller-Casseres et al., 2024. The numbers below indicate these results, but also highlight the increasing electrification of transport and the use of energy carriers such as hydrogen. For example, in some scenarios (1.5_gCCS) the use of petroleum derivatives in passenger transport reduces to less than 15% of final energy use in 2050. Noteworthy, we are reporting below the final energy demand (i.e., not the services demand) but it indicates already how much fossil fuels are displaced by fleet electrification by comparison with the NPi results. Müller-Casseres, E., Leblanc, F., van den Berg, M., Fragkos, P., Dessens, O., Naghash, H., ... & Schaeffer, R. (2024). International shipping in a world below 2° C. Nature Climate Change, 1-8.

Notes/actions

We included the following plots as Supplementary Figures, and referred to them in the manuscript as follows.

Supplementary Figure 1. Final energy use in the freight transportation sector across scenarios.

Supplementary Figure 27. Final energy use in the passenger transportation sector across scenarios.

FROM: (37% on final energy use basis considering passenger and freight transport in 2070)

	TO: (37% on final energy use basis considering passenger and freight transport in 2070, see Supplementary Figs. 16 and 17)
--	--

Comment 4	Reviewer 3#, comment 7 – You haven't addressed one of the critical points of the reviewer, about the assumption of biomass/energy carbon neutrality. You have noted that COFFEE includes land use emissions, but you stated what you assume about harvest species, harvest method, soil carbon effects, etc. All these things could lead to biomass or bioenergy being very far from carbon neutral. Just state your assumption of neutrality and the basis for it.
Response	Thank you for your comment. We apologise for missing this point in the previous review. We agree that biomass and bioenergy can induce positive emissions, especially if we consider non-CO₂ emissions. However, within the balance between carbon emissions and sinks, the model identifies and favours options and agriculture practices that guarantee net negative carbon emissions. These strategies are important sources of negative emissions needed to reach global targets. We assume carbon neutrality for biomass based on specific conditions. Firstly, the model includes emissions from land-use changes, energy use during agricultural practices and logistics, agricultural residues, fertilizer use, and other relevant sources. It also considers carbon sequestration during biomass growth and the final storage of carbon, either geologically or in materials. To address the reviewer's specific concerns:  • Harvest Species: COFFEE represents the following crops: Sugarcane, Corn, Wheat, Woody biomass, Grassy biomass, Beet, Bagasse, Residues. We consider the growth rates and carbon uptake capacities of different species to ensure that the biomass used has a high potential for carbon neutrality. • Harvest Methods: We do not explicitly represent harvest methods, but are indirectly assumed via crop yields, energy inputs, capital investment and operating costs (incl. labour) • Soil Carbon Effects: The model does not account for soil carbon changes due to biomass harvesting. Soil carbon is affected only through land-use change. From the optimization perspective, there is no reason why the model would select routes and options that: a) are above the shadow price and would increase the cost of the system; b) that would not provide net negative emissions, as this would not help in achieving the carbon budget defined and would increase the total cost of mitigation (i.e., therefore requiring more mitigation and removal in other sectors). More information about the land-use sector in COFFEE can be found in the following references:

	Köberle, A.C., Daioglou, V., Rochedo, P. et al. Can global models provide insights into regional mitigation strategies? A diagnostic model comparison study of bioenergy in Brazil. Climatic Change 170, 2 (2022). https://doi.org/10.1007/s10584-021-03236-4 Rochedo, P. R. R. (2016). Development of a global integrated energy model to evaluate the Brazilian role in climate change mitigation scenarios. DS Thesis, Universidade Federal do Rio de Janeiro, Brazil Hasegawa, T., Fujimori, S., Frank, S. et al. Land-based implications of early climate actions without global net-negative emissions. Nat Sustain 4, 1052–1059 (2021). https://doi.org/10.1038/s41893-021-00772-w https://www.iamcdocumentation.eu/index.php/Model_Documentation_-_COFFEE-TEA
Notes/actions	To address the reviewer’s concern, we included the following section in the Supplementary Methods: S1.1. Land-use emissions considerations COFFEE assumes carbon neutrality for biomass based on specific conditions. Firstly, the model includes emissions from land-use changes, energy use during agricultural practices and logistics, agricultural residues, fertilizer use, and other relevant sources. It also considers carbon sequestration during biomass growth and the final storage of carbon, either geologically or in materials. Within the balance between carbon emissions and sinks, the model identifies and favours options and agriculture practices that guarantee net negative carbon emissions. COFFEE represents Sugarcane, Corn, Wheat, Woody biomass, Grassy biomass, Beet, Bagasse, Residues. It considers the growth rates and carbon uptake capacities of different species to ensure that the biomass used has a high potential for carbon neutrality. Harvest methods are not explicitly represented, but are indirectly assumed via crop yields, energy inputs, capital investment and operating costs (incl. labour). The model does not account for soil carbon changes due to biomass harvesting. Soil carbon is affected only through land-use change. More information on the land-use sector in COFFEE can be found in refs.^{87–90}

REVIEWERS' COMMENTS

Reviewer #1 (Remarks to the Author):

I would like to thank the authors for their hard work. They have addressed all the concern. Meanwhile, I've noticed one last minor issue. Could the authors elaborate how CO₂ emission was estimated in the Methods section, including whether the authors have assumed 0 for the CO₂ emission of renewable energy?

Reviewer #2 (Remarks to the Author):

I think the revised manuscript is fine with publication.

Point-by-point response to reviewers

3rd round

Dear Editor and Reviewers,

We wish to express our gratitude for the comments and suggestions. In this letter, we respond to the points raised in the 3rd round of reviews, clearly indicating the amendments made to our updated manuscript *Unaddressed non-energy use in the chemical industry can undermine fossil fuels phase-out*.

Once again, and for the sake of clarity:

- i) Reviewers' comments are written in plain text;
- ii) Our responses are in blue;
- iii) The text quoted from the paper is green.

Reviewer #1:

I would like to thank the authors for their hard work. They have addressed all the concern. Meanwhile, I've noticed one last minor issue. Could the authors elaborate how CO₂ emission was estimated in the Methods section, including whether the authors have assumed 0 for the CO₂ emission of renewable energy?

We thank the reviewer for the positive feedback and constructive comments during the reviewing process.

We added the text below to the methods section:

GHG emissions are accounted for using the IPCC methodology, which includes emissions from fuels, industrial processes, waste, and Agriculture, Forestry, and Other Land Uses (AFOLU), covering greenhouse gases such as CO₂, N₂O, and CH₄. Emissions from biomass production (land-side) for subsequent conversion into energy carriers or materials are accounted for in the AFOLU sector. Additionally, the model incorporates options for carbon dioxide removal, such as land sinks, carbon storage associated with BECCS or DAC, and the use of materials in long-lived applications. Unlike life cycle assessments (LCA) studies that consider cradle-to-grave emissions, COFFEE focus solely on emissions directly associated with major energy, food, and industrial products. Therefore, energy renewable energy sources like solar and wind are considered to provide zero emissions electricity.

Reviewer #2:

I think the revised manuscript is fine with publication.

We thank the reviewer for the constructive comments during the reviewing process.